# A multiscale signalling network map of innate immune response in cancer reveals cell heterogeneity signatures

Maria Kondratova[1,4], Urszula Czerwinska [1,2,4], Nicolas Sompairac [1,2], Sebastian D. Amigorena[3], Vassili Soumelis [3], Emmanuel Barillot [1], Andrei Zinovyev [1] & Inna Kuperstein [1]*

The lack of integrated resources depicting the complexity of the innate immune response in cancer represents a bottleneck for high-throughput data interpretation. To address this challenge, we perform a systematic manual literature mining of molecular mechanisms governing the innate immune response in cancer and represent it as a signalling network map. The cell-type specific signalling maps of macrophages, dendritic cells, myeloid-derived suppressor cells and natural killers are constructed and integrated into a comprehensive meta map of the innate immune response in cancer. The meta-map contains 1466 chemical species as nodes connected by 1084 biochemical reactions, and it is supported by information from 820 articles. The resource helps to interpret single cell RNA-Seq data from macrophages and natural killer cells in metastatic melanoma that reveal different anti- or pro-tumor sub-populations within each cell type. Here, we report a new open source analytic platform that supports data visualisation and interpretation of tumour microenvironment activity in cancer.

[1] Institut Curie, PSL Research University, Mines Paris Tech, Inserm, U900, 75005 Paris, France. [2] Université Paris Descartes, Centre de Recherches Interdisciplinaires, Paris, France. [3] Institut Curie, PSL Research University, Inserm, U932, 75005 Paris, France. [4] These authors contributed equally: Maria Kondratova, Urszula Czerwinska. *email: inna.kuperstein@curie.fr

Tumors are engulfed in a complex microenvironment (TME) that critically impacts disease progression and response to therapy. TME includes immune and non-immune interconnected components that exchange multiple signals and are influenced by molecules secreted by cancer cells. The behavior of the tumor and its TME as a whole critically depends on the organization of these different players and their ability to regulate each other in a dynamic manner[1]. The innate immune part of the TME plays important, but sometimes opposite roles in tumor evolution. Innate immune cells can contribute to eliminate the tumor, e.g. through phagocytosis and T cell priming and by induction of adaptive immune response. However, they can also favor tumor escape from immunological control, by a production of immunosuppressive molecules such as transforming growth factor beta (TGFB) or interleukin 10 (IL10)[2]. An additional level of complexity in the TME is that various stimuli can lead to a range of innate immune cells' phenotypes. This results in very heterogeneous subpopulations within each innate immune cell type coexisting in TME[3,4].

Depending on the set of stimuli from TME and tumor, immune cells are able to change their phenotype or polarization status from anti-tumor to pro-tumor[5,6]. Such functional dichotomy was first evidenced for one of the components of innate immunity in TME, the tumor-associated macrophages (TAM) and led to a description of M1 and M2 polarized TAM classes[7]. The same tendency was later documented for other components of innate immunity as neutrophils[8], dendritic cells[9] and natural killers[10]. Therefore, the term "polarization" can be applied for the innate immunity system in TME in general[11] that represents the major focus of current works. The balance between anti-tumor and pro-tumor activity of innate immune cells has an impact on tumor growth, patient response to therapy, and survival[12].

The correct evaluation of the polarization status within the subtle innate immune cell subpopulations in TME is essential for immunotherapy improvement. Nevertheless, the primary activation of adaptive immune response requires innate immune players, the antigen-presenting cells (APC) such as dendritic cells[13] or macrophages[14,15] Therefore, an efficient immune checkpoint therapy depends directly on the proper innate immune activation[16]. In addition, there are studies showing that innate immunity can restrict tumor growth even when the adaptive immune system is inactivated[17]. This indicates that detailed study of potential innate immune-related targets should be performed to identify new types of immunotherapy[18] that could function in synergy with the current T cell-targeted therapies or act independently[19,20].

There is a massive amount of information in the literature about molecular mechanisms implicated in innate immune cells polarization in TME. However, most of the studies are focused on individual molecular components and pathways. They do not integrate the complexity of multiple crosstalks between innate immune cells and tumor cells. To create a holistic picture of the diversity and integrity of innate immune system in TME, the knowledge about molecular circuits should be gathered together and systematically represented[21].

To address these challenges, a systems biology approach is needed[22]. Formalization of biological knowledge in a form of comprehensive signaling maps, both at the intra- and intercellular levels, helps to integrate information from multiple research papers[23]. There are numerous public databases containing signaling pathways related to innate-immune response such as KEGG[24] and REACTOME[25], which are quiet comprehensive, but contain mostly generic mechanisms. Furthermore, there are resources dedicated to different types of innate immune cells such as macrophages[26] or dendritic cells[27]. Finally, there are resources depicting the innate immune system in general as InnateDB[28]

and ImmuNet[29], Virtually Immune[30]. However, these repositories are rather pathogen response-oriented than cancer-specific and often represent a catalog of disconnected pathways. Thus, there is a need to create an integrated resource on molecular mechanisms of innate immune response in cancer.

To fill the gap, we construct and present here a system of cell-type-specific maps and an integrated meta-map of innate immune signaling in cancer based on the information retrieved from the literature (Fig. 1). These maps together represent an open source analytic platform for data visualization and interpretation of TME activity in cancer.

## Results

**Principles of innate immune response in cancer.** The molecular mechanisms regulating six major innate immune cell types found in the TME were gathered and depicted in the form of network maps. To cope with a massive body of literature on innate immune response in cancer we followed a systematic procedure of literature selection, knowledge organization, and integration of information in a visual and understandable manner (Fig. 1). The network maps were constructed as two-dimensional maps to facilitate the graphical representation of molecular mechanisms that drive biological processes. The maps possess a particular layout that reflects the accepted vision of spatial organization and propagation of biological processes. The information about molecular mechanisms was manually retrieved by the researchers from the scientific literature along with the information presented in general pathway databases or in the immune system-specialized resources. The information was classified by specificity to the cell types in cancer and organized into three cell-type-specific signaling network maps, namely map of macrophages and myeloid-derived suppressor cells (MDSC), dendritic cells and natural killer (NK) cells (Fig. 2). These maps, enriched by the information on additional cell types as neutrophils and mast cells, were integrated into the meta-map of innate immune response in cancer (Fig. 3).

The molecular mechanisms were depicted in the maps in the form of biochemical reaction network using a well-established methodology[31,32]. The maps were described using Systems Biology Graphical Notation language (SBGN)[33] and drawn using the CellDesigner tool[34] that ensures compatibility of the maps with various tools for network analysis, data integration, and network modeling (Fig. 3b). Each molecular player and reaction in the maps was annotated in the NaviCell format. The NaviCell annotations include PubMed references, cross-references with other databases, and notes of the map manager. In addition, molecular complexes and reactions were assigned with confidence scores and tags indicating their involvement in different biological processes on the maps. Finally, the correspondence of each molecular player on the map to different cell types is also indicated, indicated by cell-type-specific tags (Supplementary Fig. 1)[35]. The principles and procedure for map construction are provided in the Methods.

**Content and structure of the innate immune maps.** Macrophages are the major immune component of leukocyte infiltration in the tumor. The anti-tumor polarization of macrophages is related to their ability to recognize and to reject tumor cells by phagocytosis, represent tumor antigens on the cell surface and induce a T cell response and attract immune cells into the TME. However, TAMs can also act as pro-tumor agents, expressing tumor-stimulating growth factors, producing immunosuppressive molecules induce angiogenesis and matrix remodeling in TME and consequently facilitate metastatic process[36,37].

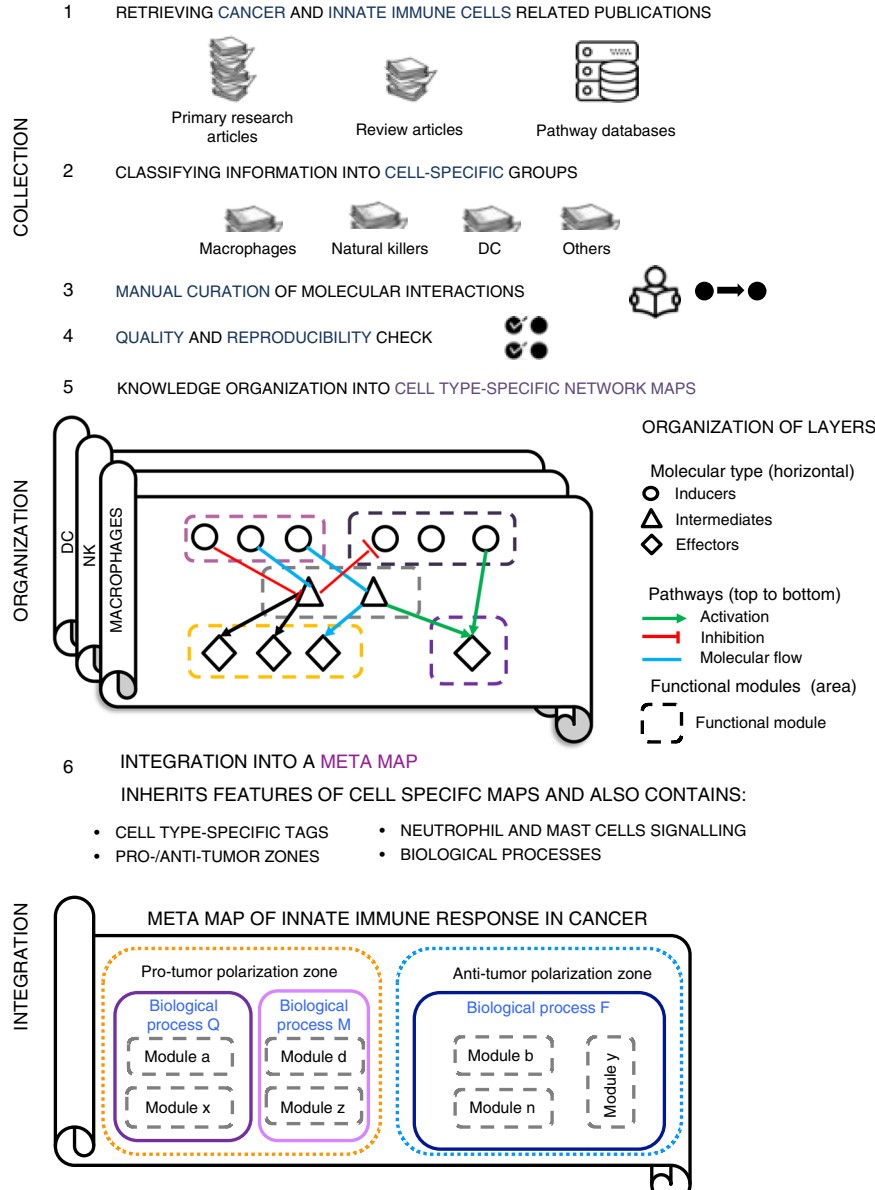

**Fig. 1** Map construction workflow and map structure. The scheme demonstrates the steps of meta-map construction starting from collection of cancer-specific and innate-immune specific information about individual molecular interactions from scientific publications and databases, manual annotation and curation of this information (steps 1–4), then organization of this formalized knowledge in form of cell-type specific maps (step 5), and finally integration the cell-type specific networks in one global meta-map of innate immune response in cancer with areas corresponding to biological processes, modules, pro- and anti-tumor polarization (step 6)

MDSC represent a heterogeneous population of myeloid cells. In general, the role of MDSC in TME is similar to TAMs. MDSC suppress T cell response and NKs' activity in TME. In addition, MDSCs induce EMT and angiogenesis and participate in matrix remodeling. MDSC mostly show a pro-tumor activity; therefore, their presence in the tumor is correlated with a poor clinical prognosis[38,39]. The MDSC signaling is included into the macrophage cell-type-specific map.

The macrophage and MDSC cell-type-specific map contains 588 objects and 7 modules representing both pro-tumor and anti-tumor polarization of myeloid cells (Fig. 2a, Supplementary Table 1).

The map is available at https://navicell.curie.fr/navicell/newtest/maps/macrophages_mdsc_cells/master/index.html.

Dendritic cells are innate immune cells that can have both myeloid and lymphoid origin. As with macrophages, dendritic cells have phagocytic abilities and can produce inflammatory cytokines. But the major role of dendritic cells in anti-tumor response is antigen presentation and further T cell activation[40]. The dendritic cell map contains 491 objects and 8 modules (Fig. 2b, Supplementary Table 1).

The map is available at https://navicell.curie.fr/navicell/newtest/maps/dendritic_cell/master/index.html.

NKs are big granular lymphocytes that can be cytotoxic to tumor cells. The main role of NK cells in innate immunity is an elimination of cells lacking MHC1 molecules that therefore cannot be recognized by T cells. The activity of NK cells is stimulated by the target cells expressing NK receptors activating ligands and modulated by inflammatory cytokines, produced by macrophages and dendritic cells. NK cells secrete granules contains lytic enzymes and express the apoptosis inducers. Presence of active NK cells in cancer is correlated with good

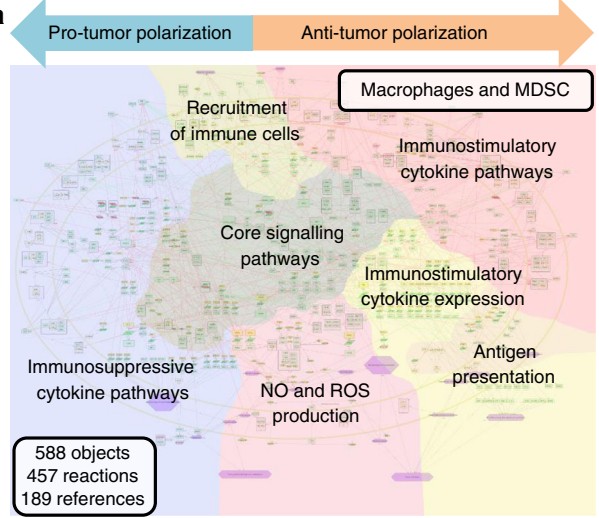

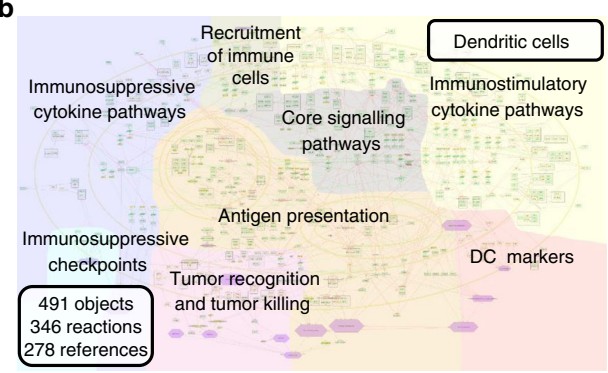

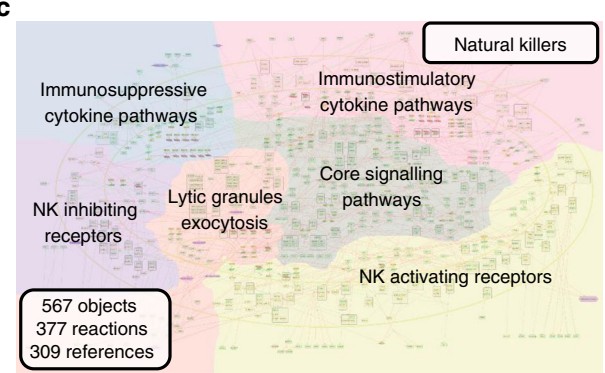

**Fig. 2** Cell-type-specific maps. Cell-type-specific networks are visualized at the top-level view, the colorful background indicates boundaries of functional modules of the maps. **a** The maps of macrophages and MDSC. **b** The map of dendritic cells. **c** The map of natural killer cells

documented, but it is known that they can produce ROS, inflammatory cytokines and demonstrated tumoricidal activity. Although, in other conditions, neutrophils act as pro-tumor agents via stimulation of matrix remodeling, angiogenesis, and metastasis, therefore these cells have both pro- and anti-tumor polarization potential[8,42]. The signaling on neutrophils is included into the innate immune meta-map (Fig. 3, Table 1).

Mast cells resemble blood basophils and contain granules rich in histamine and heparin. The experimental data about the influence of mast cells on tumor microenvironment is contradictory. It is known that mast cells can produce inflammatory cytokines and secrete Chondroitin sulfate which acts as a decoy for tumor cells and blocks the metastatic process. However, mast cells also secrete molecules stimulating tumor growth, angiogenesis and local immunosuppression[43,44]. Probably the polarization of mast cells in TME is context-dependent. The signaling on mast cells is included into the innate immune meta-map (Fig. 3, Table 1).

The aforementioned cell-type-specific maps gathered together and enriched by additional information gave rise to the global, seamless meta-map of innate immunity in cancer. The meta-map contains 1466 chemical species as nodes connected by 1084 biochemical reactions, and it is supported by information from 820 cell-type specific and cancer-related articles (Table 1).

The layout design of the meta-map reflects the current understanding of signaling propagation in cells. To cope with the complexity of the signaling network and to make it understandable and navigable, the meta-map has a hierarchical structure (Figs. 1 and 3). The meta-map possesses two major structuring dimensions: the internal organization of the map (layers, zones, meta-module, modules, and pathways) and the external organization represented by zoom levels (see explanation below).

The internal organization of the meta-map is provided in a form of three layers entitles Inducers, Core Signaling, and Effectors (Fig. 3a, Table 1). The top part of the meta-map is the Inducers layer that depicts inducer molecules frequently present in TME. The inducers interact through specific receptors and adaptor proteins that propagate the signal via limited number of transmitters, also called hub molecules as NF-kB, PLCG, PI3K, etc. These molecules are located in the middle parts of the meta-map in the Core Signaling layer. The signaling is further propagated to the Effectors layer, located in the lower part of the meta-map, which actually executes the biological activity and therefore defines the outcome phenotype, namely, the positive or negative influence of the innate immunity system on the tumor growth and invasion (Fig. 3b, Table 1).

Further, the whole meta-map is divided into multiple signaling pathways, running through the aforementioned layers (Fig. 3b). A signaling pathway on the meta-map represents a sequence of molecular interaction which transforms extracellular signals into intracellular activity or into single or multiple cell phenotypes. For instance, the TGFB pathway in innate immune cell upregulates the expression of immune-suppressive ligands, inhibits expression of immune-activating molecules and NO production, and modulates migration of immune cells (Supplementary Fig. 2A).

The meta-map is composed of 98 signaling pathways, 30 of which contain more than 10 molecules in the sequence (Supplementary Data 2). It is worth highlighting that there are many crosstalks between different signaling pathways (Fig. 3b).

The signaling pathways of the meta-map form together 25 functional modules. A module on the meta-map represents a group of signaling pathways collectively executing a phenotype, e.g. the functional module NO and ROS production contains several signaling pathways implicated in a single biological function (Supplementary Fig. 2B).

prognosis. To escape NK control, tumor cells express immunosuppressive cytokines and downregulate NK ligands expression that collectively inhibit cytotoxic activity of NK cells[41]. A pro-tumor polarization of NK cells is not described in the literature. However, suppressed NK cells are incapable to reject tumor cells and, therefore, indirectly promote cancer progression. The NK map contains 567 objects and 6 modules (Fig. 2c, Supplementary Table 1).

The map is available at https://navicell.curie.fr/navicell/newtest/maps/natural_killer_cell/master/index.html.

Neutrophils form a subtype of granulocytic leukocytes. The role of neutrophils in the tumor microenvironment is not well

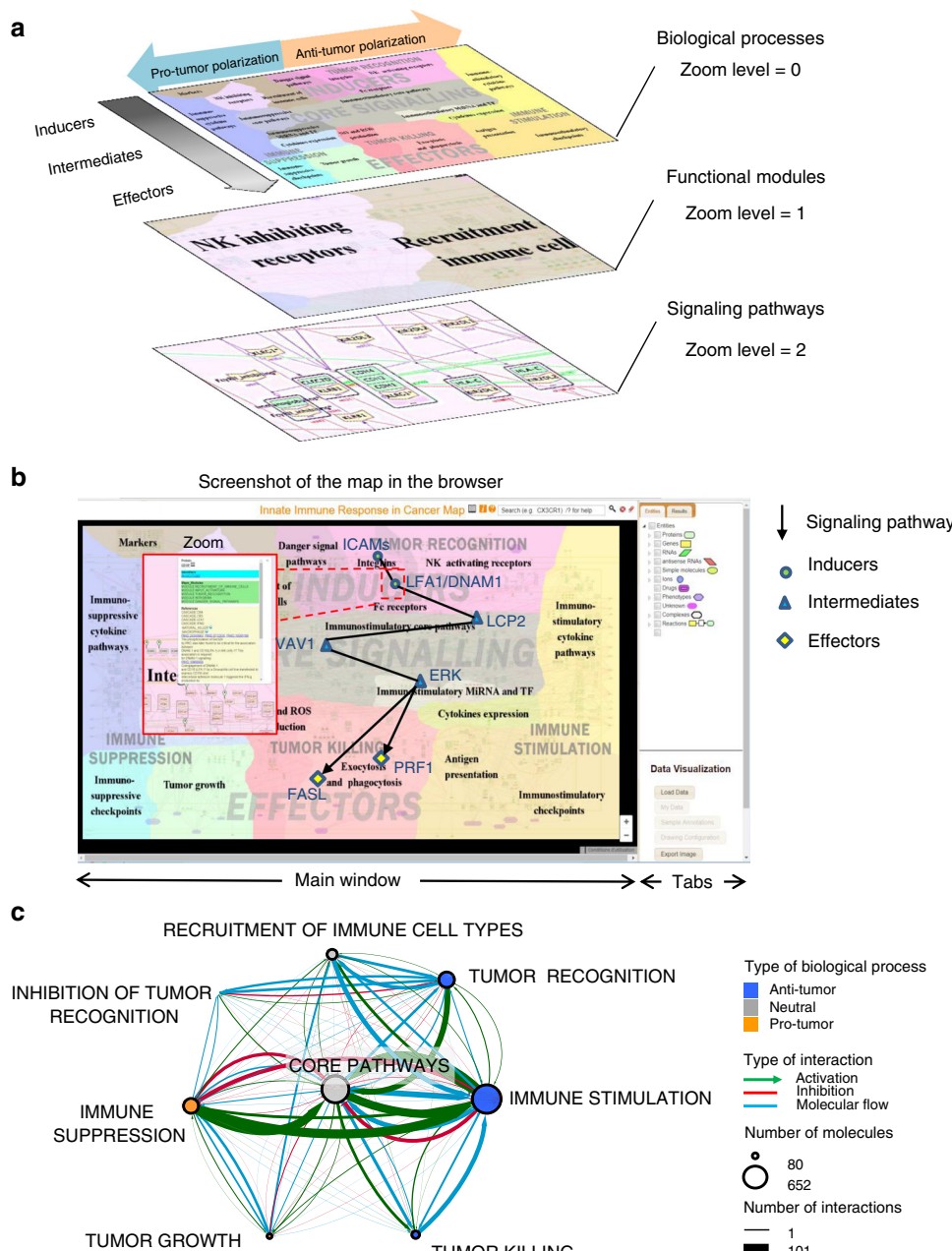

**Fig. 3** Structure of meta-map of innate immune response in cancer. **a** Top view layout of the innate immune meta-map. Functional modules represent key processes involved in pro-tumor and anti-tumor activity of innate immunity in cancer, showed at different zoom levels (0—polarization and biological processes, 1—functional modules, 2— signaling pathways, molecules, interaction types, and annotation details. **b** Signaling pathways in the meta-map structure in a browser window. **c** The network of modules demonstrating crosstalks between biological processes represented on the meta-map. Nodes represent biological processes with the size associated to number of molecules in a process, color of the node is related to pro/anti-tumor polarization (see legend), interactions reflect cross-talk between the biological processes, the thickness of the edge is related to number of interactions and the color to the nature of interactions

These functional modules are assembled into the structures of higher level, namely nine biological processes (meta-modules), reflecting the major biological activities of the innate immune system with respect to a tumor, i.e. Tumor recognition, Inhibition of Tumor Recognition, Tumor Growth, Tumor Killing, Immune Stimulation, Immune Suppression, Recruitment of Immune Cells, Core Activation, and Core Inhibition.

Finally, at the highest level, all biological processes (meta-modules) are grouped into two zones representing the concept of innate immune system polarization into anti- or pro-tumor mode. The Anti-Tumor zone covers the meta-modules named Tumor Recognition, Immune Activation, Tumor Killing, and Core Activation, whereas the Pro-Tumor zone is composed of Inhibition of Tumor Recognition, Immune Suppression, Tumor Growth and Core Inhibition meta-modules (Figs. 1, 3a and Table 1). The list of map nodes per signaling pathways, modules, biological processes (meta-modules), and zones is available in the Supplementary Data 3 and downloadable form the resource website (https://navicell.curie.fr/pages/maps_innateimmune.html).

The various map levels are interconnected and cross-talk to each other. The crosstalks between the biological processes (meta-modules) are represented as an interaction network shown

**Table 1 Hierarchical modular structure of innate immune response meta-map**

| Zones metamodule module | Chemical species as entities | Proteins | Genes | RNAs | asRNAs | Reactions | References |
|---|---|---|---|---|---|---|---|
| **Zone: Pro-tumor polarization** | | | | | | | |
| Inhibition of Tumor Recognition | | | | | | | |
| NK inhibiting receptors | 35 | 23 | 1 | 1 | 0 | 14 | 57 |
| Immune Suppression | | | | | | | |
| Immunosuppressive cytokine pathways | 109 | 46 | 10 | 11 | 3 | 67 | 114 |
| Immunosuppressive cytokine expression | 55 | 19 | 14 | 14 | 0 | 36 | 75 |
| Immunosuppressive chekpoints | 8 | 7 | 0 | 0 | 0 | 8 | 13 |
| Core Signaling Pathways | | | | | | | |
| Immunosupppressive core pathways | 43 | 23 | 5 | 5 | 1 | 25 | 54 |
| MIRNA TF Immunosuppressive | 77 | 20 | 23 | 14 | 12 | 48 | 62 |
| Tumor Growth | | | | | | | |
| Tumor growth | 60 | 42 | 8 | 8 | 0 | 71 | 58 |
| **Zone: Anti-tumor polarization** | | | | | | | |
| Tumor Recogntiton | | | | | | | |
| NK activating receptors | 114 | 45 | 16 | 14 | 6 | 72 | 115 |
| Danger signal pathways | 60 | 30 | 2 | 1 | 0 | 36 | 66 |
| FC receptors | 18 | 12 | 0 | 0 | 0 | 8 | 37 |
| Integrins | 38 | 24 | 0 | 0 | 0 | 21 | 56 |
| Immune Stimulation | | | | | | | |
| Immunostimulatory cytokine pathways | 152 | 74 | 18 | 18 | 3 | 92 | 193 |
| Immunostimulatory cytokine expression | 43 | 17 | 12 | 11 | 1 | 27 | 109 |
| Antigen presentation and immunostimulatory checkpoints | 99 | 65 | 6 | 6 | 0 | 91 | 152 |
| Core Signaling Pathways | | | | | | | |
| Immunostimulatory core pathways | 184 | 93 | 6 | 6 | 114 | | 244 |
| MIRNA TF immunostimulatory | 50 | 17 | 12 | 10 | 5 | 33 | 60 |
| Tumor Killing | | | | | | | |
| Lytic granules exocytosis and phagocytosis | 73 | 39 | 6 | 6 | 5 | 50 | 75 |
| No ROS production | 33 | 10 | 4 | 4 | 0 | 23 | 44 |
| **Cell-type specific markers** | | | | | | | |
| Markers | | | | | | | |
| Markers macrophage | 22 | 10 | 6 | 6 | 0 | 0 | 8 |
| Markers NK | 10 | 10 | 0 | 0 | 0 | 0 | 36 |
| Markers mast | 6 | 6 | 0 | 0 | 0 | 0 | 9 |
| Markers DC | 16 | 14 | 0 | 2 | 0 | 0 | 14 |
| Markers neutrophil | 11 | 11 | 0 | 0 | 0 | 0 | 15 |
| Markers MDSC | 9 | 9 | 0 | 0 | 0 | 0 | 9 |
| **Recruitment** | | | | | | | |
| Recruitment of immune cells | | | | | | | |
| Recruitment of immune cells | 103 | 48 | 17 | 17 | 0 | 93 | 83 |
| Meta-map | 1466 | 582 | 162 | 152 | 20 | 1084 | 820 |

Structure and content of innate immune meta-map

in the Fig. 3c. The interaction network demonstrates different types of links between meta-modules of the map, including activation, inhibition, molecular flow. The Core Signaling meta-module is a network "hub" where most signaling pathways converge. In addition, it is notable that there are numerous positive and negative crosstalks between Immune Stimulation and Immune Suppression meta-modules on the map (Fig. 3c).

The external organization of the meta-map is reflected in the hierarchical structure of zoom levels, similar to geographical maps, where only limited information is displayed on each zoom level (Fig. 3a). This hierarchical structure facilitates Google Maps-like navigation of the map.

**Access, navigation, and maintenance of the resource**. The cell-type-specific and the integrated meta-map are open source, can be browsed online, and are available at https://navicell.curie.fr/pages/maps_innateimmune.html. Each map is presented under three independent platforms, namely NaviCell, MINERVA, and NDEx. All map components are clickable, making the map interactive. The extended annotations of map components contain rich tagging system converted to links and confidence scores.

This allows tracing the involvement of molecules into different map sub-structures as pathways, modules, and biological processes (meta-modules) (Fig. 3). Tagging system also allows to use the meta-map as a source of annotated signatures (Supplementary Fig. 1).

The semantic zooming feature of NaviCell[35] simplifies the navigation through large maps of molecular interactions, showing readable amount of details at each zoom level.

**Comparison of meta-map with existing pathway databases**. The meta-map content (Supplementary Fig. 3) and the coverage of literature used to annotate the entities (Supplementary Fig. 4) were compared to a sub-set of pathways related to the innate immune system from the existing molecular interaction databases (Supplementary Table 2). The InnateDB database contains a detailed description of the innate-immune signaling, even though more general databases as KEGG and REACTOME also include immune pathways. A description of comparison procedure is provided in the Methods.

We further compared the major features of innate immune response representation in different pathway databases. The

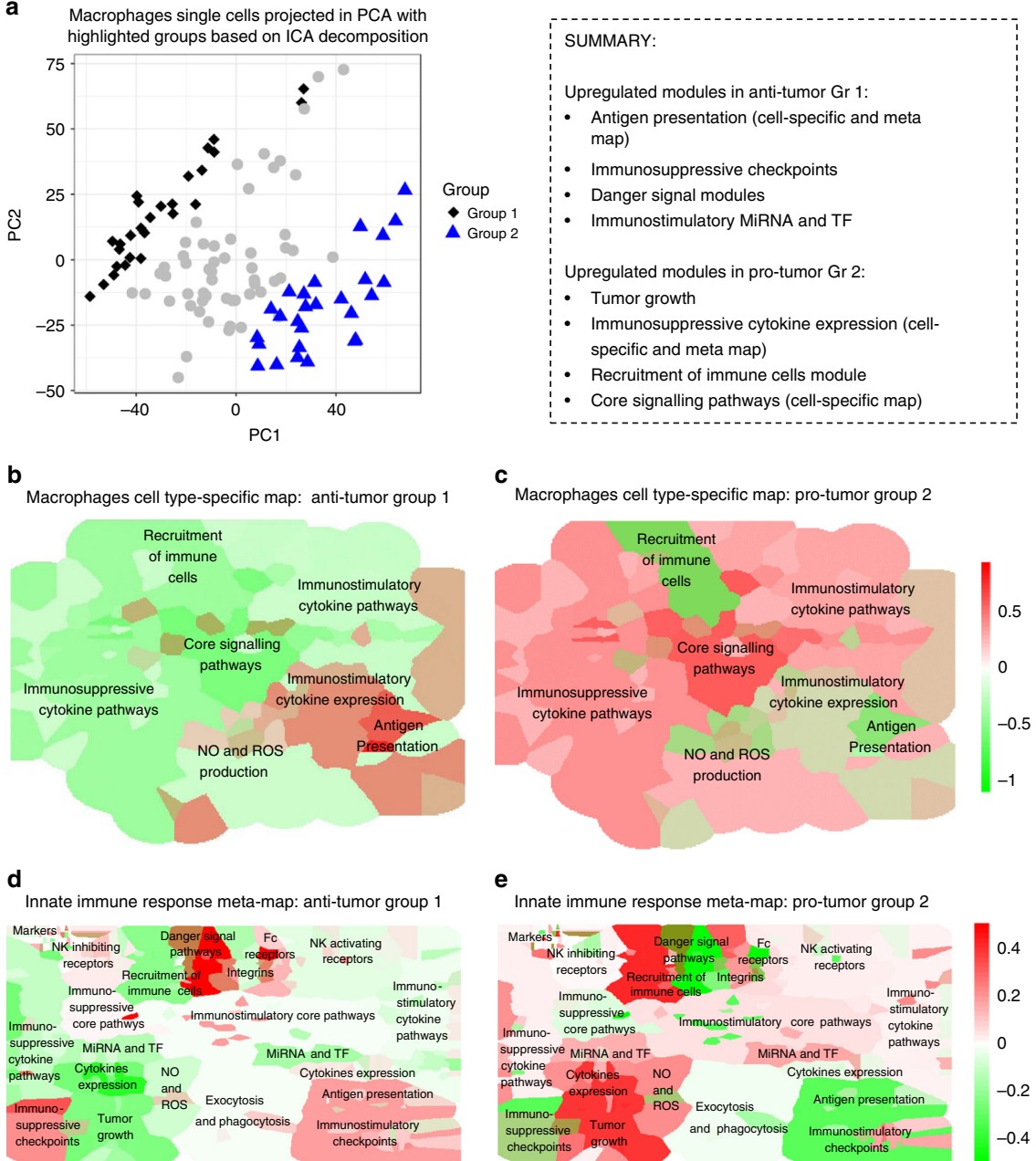

**Fig. 4** Visualization of modules activity scores using expression data from melanoma macrophages. **a** Macrophages single cells in PC1 and PC2 coordinates space. Two groups, the first and the fourth quartiles of distribution along the IC1 axis, are colored distinctly in blue and black. Staining of the macrophage cell-type-specific map with modules activity scores calculated from single-cell RNAseq expression data for **b** Macrophages group 1 (Anti-tumor) and **c** Macrophages Group 2 (pro-tumor) cells. Staining of the innate immune meta-map with modules activity scores calculated from single-cell RNAseq expression data for **d**. Macrophages Group 1 (Anti-tumor) and **e** Macrophages Group 2 (Pro-tumor) cells. Color code: red—upregulated, green—downregulated module activity

innate immune response in cancer resource contains cell-type-specific maps in contrast to other databases. The comparison indicates that the cross-talk between the pathways is visually represented at the maps of immune response in cancer resource. Finally, the combination of hierarchical organization of knowledge and possibility of navigation through the layers of the maps thanks to semantic zooming feature makes the innate immune resource more suitable for meaningful data visualization. The visualization tool box is built into the NaviCell environment which allows easy data integration and visualization in the context of the innate immune maps (Figs. 4 and 5).

Taken together, the results of database comparisons indicate that the innate immune response in cancer resource is topic-specific, and describes immune-related and cancer-relevant signaling processes based on the latest publications about innate immune component in TME. The thoughtful layout and visual organization of the biological knowledge on the maps makes it a distinguished resource for data analysis and interpretation.

**Application of the maps for data visualization and analysis.** The cell-type-specific maps and the meta-map were applied to explore the heterogeneity of innate immune cell types in cancer.

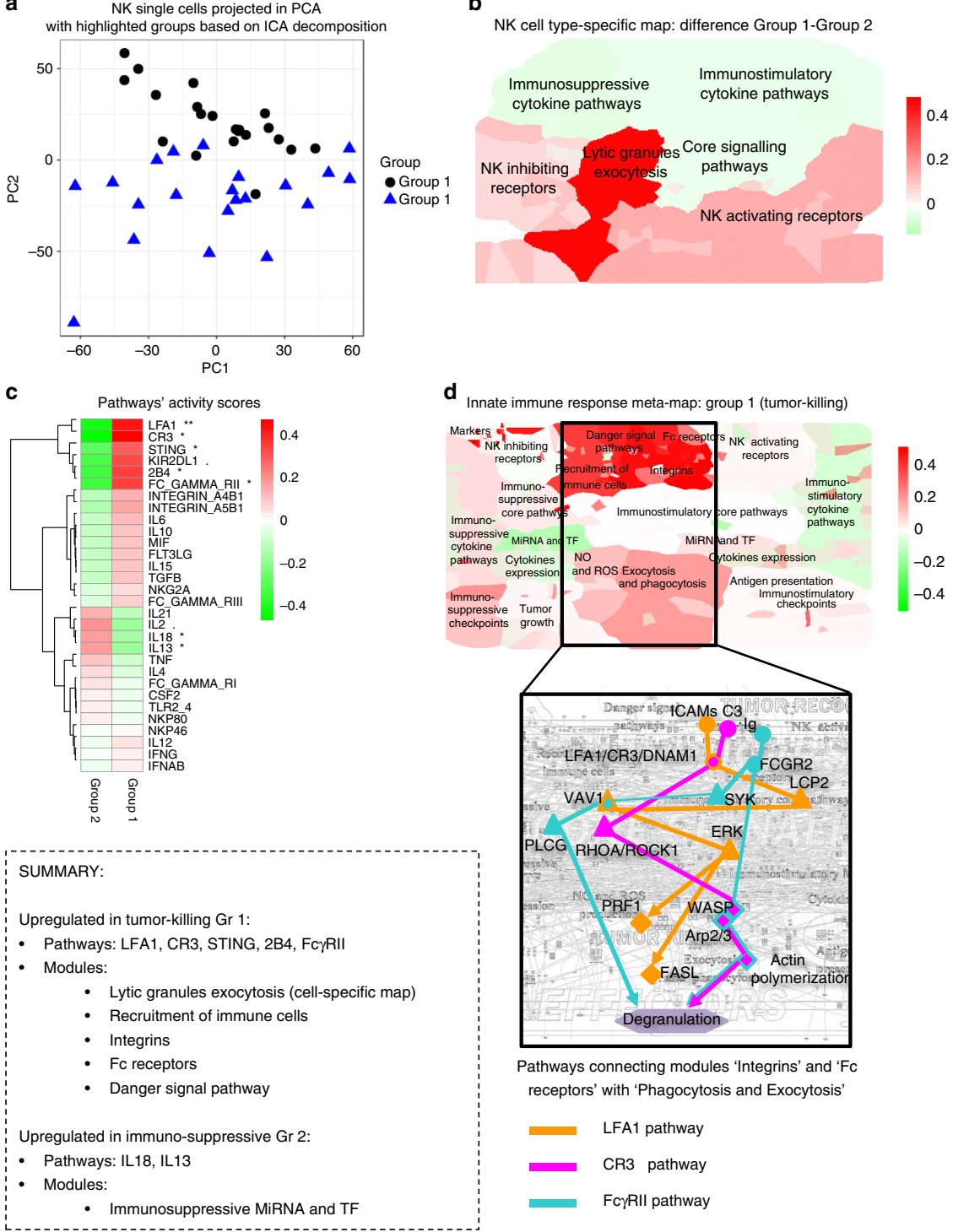

**Fig. 5** Visualization of modules activity scores using expression data from melanoma natural killers (NK). NK single cells in PC1 and PC2 coordinates space. Two groups are colored distinctly in blue and black. **a** Map staining of the NK cell-type-specific map with modules activity scores calculated from single-cell RNAseq expression data for **b** NK Group 1. **c** Heatmap of activity scores in signaling pathways of NK groups. Map staining of the innate immune response meta-map with modules activity scores for **d** NK Group 1 ("tumor killing") with a zoom into three signaling pathways relating the two upregulated modules: "Danger signal pathways" and "Exocytosis and phagocytosis" with main molecular players named. Color code: red–upregulated, green—downregulated module activity. The *p* values of the *t* test were reported in the heatmaps with the standard code of significance (***$p < 0.001$, **$p < 0.01$, *$p < 0.05$, < 0.1)

The single-cell RNA-Seq data for macrophages and NK cells from metastatic melanoma samples were used[45].

A matrix factorization technique, independent components analysis (ICA)[46] allows ranking genes or samples along data-driven axes. The independent components instead of detecting highest variability axes as PCA, extract independent and non-Gaussian signals called components. The most stable component was used as a way to order the cells based on some latent process that we aim to interpret using innate immune maps. In order to better understand the differences in the cell ranking, the cells with

extreme rank values were selected, which resulted in Groups 1 and 2. When projected in the PCA space (Fig. 4a), those macrophage cell groups are lying on the borders of the cloud of points.

Furthermore, the activity scores were computed for each macrophage cell group (as defined in the Methods) for functional modules at different levels: pro- and anti-tumor general classification, innate map modules, and macrophage-specific map modules.

First, the analysis of potential pro- and anti-tumor properties of the macrophage cell groups was examined in the context of the innate immunity meta-map. Group 1 has significantly higher anti-tumor score ($t$-test $p$ value: 0.02) and Group 2 is the pro-tumor one ($t$-test $p$ value: 0.003). Second, the expression profile differences of the cells from the two groups were interpreted in the context of the Macrophage cell-type-specific map and the innate immune response meta-map. The results of the enrichment study for the two Macrophage groups were also represented as heatmaps with a significance level of $p$ value for Student's $t$-test (see Methods) (Supplementary Fig. 5). The module activity values were plotted on the maps using BiNoM plugin of Cytoscape[47].

Visualization of the module activity scores in the context of macrophage cell-type-specific demonstrates that the module Antigen Presentation is upregulated in Macrophage Group 1 (Fig. 4b) comparing to Macrophage Group 2 (Fig. 4c). Whereas, Macrophage Group 2 (Fig. 4c) shows upregulated modules Core Signaling Pathways and Immunosuppressive Cytokines Pathways comparing to Macrophage Group 1 (Fig. 4b).

Then, the module activity scores for the two Macrophage cell groups were analyzed in the context of the meta-map that allowed to detect several additional modules differentially regulated between the two groups. The four modules Antigen Presentation, Immunosuppressive Checkpoints, Danger Signal Module, and Immunostimulatory MiRNA and TF were significantly over-expressed in Anti-tumor Macrophage Group 1 ($t$-test $p$ values, respectively: $<10^{-4}$, 0.009, $<10^{-8}$, $<10^{-5}$, Fig. 4d) compared to Pro-tumor Macrophage Group 2 (Fig. 4e). On the contrary, the three modules Recruitment of Immune Cells Module, Tumor Growth, and Immunosuppressive Cytokine Expression were strongly upregulated in Pro-tumor Macrophage Group 2 ($t$-test $p$ values, respectively: $<10^{-6}$, $<10^{-6}$, $<10^{-5}$, Fig. 5d). in comparison to Anti-tumor Macrophage Group 1 (Fig. 4d, e).

From these results, it can be concluded that the Macrophage Group 1 has a tendency to express an anti-tumor phenotype, because it is characterized by the expression of inflammatory cytokines that are able to induce local adaptive immunity via antigen presentation process. Interestingly, the most typical modules responsible for tumor elimination as Exocytosis and Phagocytosis and Immunostimulatory Cytokine Pathways are not over-activated in this cell sub-set. In contrary, Macrophage Group 2 demonstrated a pro-tumor phenotype, characterized by expression of immunosuppressive cytokines restricting local immune response and growth factors supporting tumor growth.

Alike macrophages, NK cells were ranked along a latent variable obtained with ICA algorithm. Due to low cell number available, the 42 single NK cells were split in half according to the ICA ranks. Subsequently, the module activity scores were computed of each group and then a $t$-test was applied to evaluate the difference in module activity between the two NK subpopulations (Group 1 referred to as Tumor Killing and Group 2 referred to as Immunosuppressed) (Fig. 5a, Supplementary Fig. 6).

First, the comparison and visualization of the module activity between the two NK cells groups demonstrated the activation of Lytic Granules Exocytosis module in NK Group 1 compared to NK Group 2 ($t$-test $p$ value: 0.006), on the NK cell-type-specific map (Fig. 5b). The activity of this module is directly responsible

of tumor killing capacity of NK Group 1 cells that most probably exposes stronger anti-tumor abilities compared to Group 2 (Supplementary Fig. 6A).

Next, the two NK cells groups were analyzed in the context of the meta-map that allowed detection of five differentially regulated modules between the two groups of NK cells (Fig. 5d). The four modules Recruitment of Immune Cells, Integrins, Fc Receptors, and Danger Signal Pathway were significantly upregulated in the NK Group 1 comparing to the NK Group 2 ($t$-test $p$ values, respectively: 0.0001, $<10^{-4}$, 0.004, $<10^{-5}$). In contrary, the module Immunosuppressive MiRNA and TF was inhibited in the NK Group 1 comparing to the NK Group 2 ($t$-test $p$ value: 0.001). Finally, although the activity of Phagocytosis and Exocytosis module is not significantly different between the two groups, this module is rather activated in the NK Group 1 compared to the NK Group 2 (Supplementary Fig. 6B).

Collectively these results demonstrate that the NK Group 1 is characterized by upregulation of biological functions related to NK cell recruitment and activation, coinciding with upregulation of the mechanisms responsible for tumor killing. Thus, the NK Group 1 can be interpreted as newly recruited, actively migrating NKs with strong anti-tumor polarization. In contrary, most probably, NK Group 2 contains resting or suppressed NK cells that do not expose a well-defined phenotype.

The activation of upstream map zones and downstream effector zones in NK Group 1 is notable (Fig. 5d). However, which mechanisms coordinate this co-activation is not clear. The structure of the network was analyzed to address this question and the signaling pathways connecting the two activated zones were retrieved. The activation state of 30 signaling pathways from the meta-map was assessed for the cell from Group 1 and Group 2 (Fig. 5c). There are all together seven differentially regulated pathways between the two cell groups. Five are upregulated pathways in Group 1 (LFA1, CR3, STING, 2B4, FcγRII) and two upregulated pathways in Group 2 (IL13, IL18) ($t$-test $p$ values <0.05).

Within the pathways activated in the Group 1 there are three pathways regulated through receptors LFA1, CR3, and FcγRII. The key players of the pathways are presented schematically in Fig. 5d. The meta-map described difference between NK subtypes both on the level of functional modules and signaling pathways. It allows us to draw the conclusion that tumor recognition via these pathways plays an even more important role for NK-activation than well studied activation via classical NK receptors.

**Meta-map as a source of patient survival signatures**. To study whether the innate immune response meta-map can be used for assessment of processes contributing to patient survival, the list of genes from the map was used to find correlation with prognosis of patient survival using data published elsewhere[48] (see Methods). First, the presence of the genes on the innate immune response meta-map correlating with the patient survival from the aforementioned study was verified. It was detected that out of 627 proteins and protein coding genes depicted on the meta-map, 295 are significantly correlated with patient survival ($z$-score $p$ value < 0.05), that represents 47% of the map content vs. 27% in the whole genome study[48] (Supplementary Data 1).

The genes enriched on the meta-map can be divided into two groups, positively and negatively correlated with the patient survival, which confirms the observation that innate immune system can play a dual role in cancer disease. Interestingly, from the whole genome analysis in the original study by Gentles et al. (2015)[48], it emerges that there is quasi equal proportion of positively and negatively correlated genes. However, in the innate immune response meta-map, there is a strong predominance of genes positively correlated with patient survival (Table 2).

**Table 2 Distribution of genes with positive (z < 0) and negative (z > 0) correlation with patient survival across functional meta-modules in innate immune response meta-map**

| Innate immune map meta-module | Mean z-score | Positive correlation with patient survival | Negative correlation with patient survival |
|---|---|---|---|
| Tumor Growth | 1.3 | 12 | 26 |
| Inhibition of Tumor Recognition | −1.86 | 18 | 6 |
| Tumor Recogntiton | −1.56 | 67 | 28 |
| Recruitment of Immune Cells | −0.94 | 29 | 14 |
| Immune Stimulation | −0.53 | 122 | 87 |
| Tumor Killing | −0.5 | 25 | 29 |
| Core Signaling Pathways | −0.46 | 114 | 84 |
| Immune Suppression | −0.33 | 39 | 24 |

Values indicate number of genes

In order to highlight what biological functions on the innate immune response in cancer meta-map are associated to positive or negative patient survival, mean values of gene z-scores per meta-modules were calculated and visualized in the context of the meta-map (see Methods). As a general trend, the meta-map layers Inducers and Core Signaling are more significantly correlated with patient survival, compared to the layer Effectors. Furthermore, the meta-modules with biological functions related to anti-tumor activity as Immune Response Stimulation and Tumor Recognition, Recruitment of Immune Cells, etc. are positively correlated with patient survival. Interestingly the meta-module Tumor Killing is also positively correlated with patient survival, though not reaching the statistical significance (Table 2, Supplementary Fig. 7). The minority of meta-modules related to pro-tumor activity as Tumor Growth, Immunosuppressive Core Pathways, Immunosuppressive MiRNA and TF correlated negatively with patient survival (Table 2, Supplementary Fig. 7). The described analysis demonstrates that the meta-map can serve for evaluation of innate immune response signatures associated with patient survival in cancer.

## Discussion

One of the challenges of cancer biology today is understanding the phenomena of tumor heterogeneity. It consists of two relatively independent parts: first, heterogeneity of the tumor cells themselves, as a result of their clonal divergence or action of epigenetic mechanisms; second, heterogeneity of tumor micro-environment (TME). Recent years discoveries have shown that understanding how the components of this multicellular TME system interact with each other is very important for effective drug design. Actually, the attempt to modulate the interactions within the tumor microenvironment lies on the basis of new anti-cancer immune checkpoint inhibition therapy.

The analysis of large amounts of scientific information and the creation of optimal forms of its representation, require the development of new approaches for network map construction and annotation. Our first goal was to preserve the natural multidimensionality of the biological knowledge available for the different cell types in the innate component of the TME. Indeed, different cells types in innate immune system are studied from different angles. Some signaling pathways are described in detail for the macrophages and others for natural killer cells and so on. It is clear that the molecular knowledge described for one cell-type cannot always be extrapolated to another. This motivated us to create two complementary representations of innate immune system in cancer, one in the form of cell-type-specific maps and the second as an integrated meta-map of innate immune response in cancer. To be able to trace the correspondence of molecular entities and processes to a particular cell type, we introduced a

system of cell-type-specific tags, included into the annotation of all entities on the maps.

Our second goal was to provide a complete and not controversial picture on the processes occurring in the TME. The generation of an integrated meta-map of innate immunity immediately exposed a problem of map complexity. We coped with the complexity problem by introducing the hierarchical structure into the integrated meta-map, respecting the biological functions. The general layout of the integrated meta-map is based on the idea of immune cells polarization in TME, reflected in the representation of both, pro-tumor and anti-tumor signaling mechanisms. In accordance with the literature, all functional modules and meta-modules on the map are grouped into pro-tumor and anti-tumor zones. There two types of signaling modes lead to the corresponding phenotypes. In addition, the mechanism responsible for a switch in the polarization state is also represented.

The modular hierarchical map structure and complex tagging system of maps entities facilitated the production of geographical-like easily browsable open source repository. Taking an advantage of NaviCell platform, which provides Google Maps-engine and map navigation features, the innate immune maps can be explored in an intuitive way, allowing the shuttling between the cell-type-specific maps to the integrated meta-map.

NaviCell-based representation of the maps facilitates visualization of various types of omics data. Analysis of data in the context of both, cell-type-specific and integrated maps, can help in the formalization of biological hypotheses for the processes and interactions that are studied in some cell types, but unexplored in others. In addition, thanks to the rich system of tags, the maps content can be used as a source of knowledge-based gene signatures of innate immune cell type. Finally, hierarchical organization of the map provides a basis for structural network analysis, complexity reduction, and eventual transformation of the map into executable mathematical models.

The integration of the innate immune response in cancer resource into additional platforms allows broader exposure and use of the valuable maps. Therefore, in addition to NaviCell platform, the resource is also exposed in the MINERVA platform and integrated into the NDEx repository and platform. In the future, the resource will be also integrated into larger pathway collections. These moves will allow a deeper involvement of the scientific community into the maintenance and update of the maps with the latest discoveries.

The resource of innate immune maps is useful for computing network-based molecular signatures of innate immune cells polarization. These signatures will help to characterize the overall status of the signaling dictating pro-tumor and anti-tumor states of TME in cell lines and tumoral samples. It will also help to stratify cancer patients according to the status of the TME and

potentially predict patient survival and response to immunotherapies. In addition, the resource might potentially provide new immunotherapy targets, among innate immunity components of TME in tumor infiltrates. These targets can be complementary or synergistic to the well-known immune checkpoint inhibitors.

As other studies show, similar resources are used for omics data visualization in the context maps that can provide network-based molecular portraits of studied cases. Comprehensive maps are rich in molecular details carefully compiled together, therefore structural analysis of the maps can explain particular phenotypes, redundancies, and robustness[49,50]. Such analysis together with omics data can guide to design of complex druggable interventions[51]. Further, complex maps contain modules that correspond to particular biological processes; therefore, the content of these modules are used as signatures of the corresponding biological functions[52]. These lists of genes are frequently used for enrichment studies[53].

Construction of the innate immune response map is the first step in the attempt to build a global network describing the molecular interactions in the TME. The next perspective is to represent the knowledge on adaptive immune response and non-immune components in the tumor environment, including fibroblasts and endothelial cells. The final goal is to build a complete map of signaling in cancer representing both intracellular interactions of tumor cells and each component in the TME and their intracellular interactions, and describing the coordination among the components of this multicellular system.

In addition, being included into a broader Disease Maps project, the meta-map of innate immune response will be helpful, together with maps or other diseases, in the study of disease comorbidities and drug repositioning[54,55].

## Methods

**Map and model**. The maps sere drawn in CellDesigner diagram editor[34] using Process Description (PD) dialect of Systems Biology Graphical Notation (SBGN) syntax which is based on the Systems Biology Markup Language (SBML)[33]. The data model used includes the following molecular objects: proteins, genes, RNAs, antisense RNAs, simple molecules, ions, drugs, phenotypes, complexes. These objects can play the role of reactants, products, and regulators in a connected reaction network. The objects phenotypes play a role biological process outcome or readout (e.g. Migration, Tumor killing, ROS production, etc). Edges on the maps represent biochemical reactions or reaction regulations of various types. Different reaction types represent post-translational modifications, translation, transcription, complex formation or dissociation, transport, degradation and so on. Reaction regulations include catalysis, inhibition, modulation, trigger and physical stimulation. The naming system of the maps is based on HUGO identifiers for genes, proteins, RNAs and antisense RNAs and CAS identifiers for drugs, small molecules, and ions.

**Manual literature mining**. The molecular interactions reported in the scientific articles were manually curated and the information extracted from the papers was used for reconstruction and annotation of the maps. Three types of articles were used for map annotation: (i) experimental innate-immunity specific articles directly or indirectly confirming molecular interactions based on mammalian experimental data; (ii) review articles; (iii) experimental articles from non-immune cells that helped to complement the mechanisms present in immune cells (3% of the literature used for the map). In addition, pathway databases were used to retrieve information of the canonical pathways reported for the innate immune signaling general pathway databases (e.g. KEGG, REACTOME, SPIKE SignaLink, EndoNET) or in the immune system-specialized resources such as VirtuallyImmune (http://www.virtuallyimmune.org) and InnateDB (www.innatedb.com).

**Map structure and tag×ging system**. The annotation of each molecular object on the maps (protein, gene, RNA, small molecule, etc.) includes several tags indicating participation of the object in signaling pathways (tag PATHWAY:NAME), functional modules (tag MODULE:NAME), and cell-type-specific map (tag: MAP:NAME). Each PATHWAY obtains the name of the initiating ligand or receptor, in case when several ligands are acting through the same receptor. The tags are converted into the links by the NaviCell factory in the process of online map version generation. The links allow to trace participation of entities in different cell-type-specific maps and the sub-structure of the same map (pathway, module, biological process) and also facilitate shuttling between these structures.

**Reaction and protein complex confidence scores**. To provide information on the reliability of the depicted molecular interactions, two confidence scores have been introduced. Both scores represent integer numbers varying from 0 (undefined confidence) to 5 (high confidence). The reference score (REF) indicates both the number and the "weight" associated with publications found in the annotation of a given reaction. The functional proximity score (FUNC) is computed based on the external network of protein–protein interactions (PPI), InnateDB, which contains both experimental and literature-based curated interaction data[28]. The score reflects an average distance in the PPI graph between all proteins participating in the reaction (reactants, products, or regulators).

**Map entity annotation in NaviCell format**. The annotation panel followed the NaviCell annotation format of the maps includes sections Identifiers, Maps_Modules, References, and Confidence as detailed in ref. [32]. Identifiers section provides standard identifiers and links to the corresponding entity descriptions in HGNC, UniProt, Entrez, SBO, GeneCards, and cross-references in REACTOME, KEGG, Wiki Pathways, and other databases. Maps_Modules section includes tags of modules, meta-modules, and cell-type-specific maps in which the entity is implicated (see above). References section contains links to related publications. Each entity annotation is represented as a post with extended information on the entity.

**Generation of NaviCell map with NaviCell factory**. CellDesigner map annotated in the NaviCell format is converted into the NaviCell web-based front-end, which is a set of html pages with integrated JavaScript code that can be launched in a web browser for online use. HUGO identifiers in the annotation form allow using NaviCell tool for visualization of omics data. A detailed guide of using the NaviCell factory embedded in the BiNoM Cytoscape plugin[47] is provided at https://navicell.curie.fr/doc/NaviCellMapperAdminGuide.pdf.

**Depositing maps at several web-based platforms**. Cell-type specific maps and the meta-map of innate immune response in cancer were made available at other platforms such as MINERVA and NDEx. To integrate maps within NDEx, Cell-Designer maps were first loaded in Cytoscape using the BiNoM Cytoscape plugin and then uploaded on NDEx using the CyNDEx Cytoscape plugin.

**Databases content comparison**. Pathways related to the human innate immune system were selected from the InnateDB 5.4 version, except Complement Cascade (Human), NOD-like Receptor Signaling Pathway, Regulation of Autophagy (Human), and RIG-I-Like Receptor Signaling Pathway (Human). The excluded pathways represent virus and bacterial infection-specific pathways that do not correspond to TME signaling. The innate immune-related pathways from KEGG 84.1 version were retrieved from the list 5.1-Immune System. The pathways obtained from REACTOME 63rd version cover Class I MHC Mediated Antigen Processing & Presentation, MHC Class II Antigen Presentation from Adaptive Immune Branch, and all pathways from Innate Immune Branch. All together 666 gene names from InnateDB 5.4, 563 gene names from KEGG 84.1, and 2156 gene names from REACTOME 63rd were selected. These lists were compared with the innate immune response meta-map that contains 683 gene names. The complete list of selected pathways with gene names is available in the Supplementary Data 2).

The selected InnateDB pathways contain altogether, nearly the same number of objects as the innate immune response meta-map (Supplementary Data 4). The content of selected KEGG or REACTOME pathways is richer than in the innate immune response meta-map, due to the fact that KEGG and REACTOME are generic databases, describing all innate immune-related interactions, whereas the meta-maps is rather oriented to cancer signaling. The overlap between the meta-map and the three selected databases represents 61% for InnateDB, 58% for KEGG, and 30% for REACTOME. It is important to note that there are 188 genes that present exclusively at the innate immune response meta-map (Supplementary Fig. 4A, Supplementary Data 2). These unique genes are relatively homogeneously distributed across the meta-map, indicating that the depicted processes are described in more depth on the meta-map compared to the other three databases (Supplementary Fig. 3A). Several modules are significantly enriched by unique genes on the meta-map (Supplementary Fig. 3B). Thus, the modules Tumor Growth and Immunosuppressive Checkpoints contain signaling that are very well studied in cancer cells and therefore represented in great details on the meta-map. Two additional modules, entitled MIRNA TF Immunostimulatory and MIRNA TF Immunosuppressive, contain the latest information of miRNA involvement in the innate immune system control in cancer and unique for the meta-map, compared to other databases. It was concluded that the content of the meta-map is not redundant with the other pathway databases and that several functional modules directly related to TME functions are unique to the meta-map.

**Databases annotation literature comparison**. In addition, the sets of publications used to annotate the InnateDB resource and aforementioned preselected

pathway from REACTOME resource were compared to the set of publications used in the meta-map. The overlap of the literature body from the meta-map with references from InnateDB and REACTOME databases is relatively small, because 785 papers out of 820 papers that were used to annotate the meta-map are unique (Supplementary Fig. 4B). It confirms that the meta-map is not a mechanical compilation of existing databases, but rather an independent resource. It formalizes the part of biological knowledge which was not annotated before and highlights the difference between reconstruction of generic and cell-type specific pathways in terms of literature sources.

Although the median age of the literature references in the meta-map is only one year-younger compared to InnateDB and REACTOME, there is a 27% of papers dating 2010–2017 in the literature body annotating the meta-map. The literature set in the meta-map contains more papers published after year 2010 than in InnateDB and REACTOME, indicating that the meta-map represents the most recent discoveries in the corresponding fields (Supplementary Fig. 4C).

Finally, the journal types represented in the three databases were also compared. The choice of the journals used for annotating the meta-map and the other two databases is similar; however, the distribution of the papers from different types of journals is not even. The annotations of meta-map mainly contain papers from immunological journals such as *Journal of Immunology*, *Immunity*, *Nature Immunology*, and cancer-specific journals, such as *Cancer Research* and *Oncogene*, comparing to the other two databases. The annotations of InnateDB and REACTOME are rather oriented towards more generic molecular biology journals as *JBC*, *MCB*, *Nature*, and *PNAS* (Supplementary Fig. 4C and D).

**High-throughput data analytical pipeline**. Normalized melanoma data sets from GEO (GSE72056)[45] were transformed into log expression levels and mean centered. The exploratory analysis and statistical testing was performed and visualized using R packages (ggplot2, stats, pheatmap)[56–58] then MATLAB ICA implementation of FastICA algorithm[46] and icasso package[59] to improve the stability. Colored map images were obtained using function "Stain CellDesigner map" from BiNoM Cytoscape plugin[47] using .xml map files and the mean expression from the analysis described below.

**Analytical pipeline**. The single-cell molecular profiles are characterized by high variability that have both biological and technical origin. A common practice is to group single cells in order to make an aggregated representative profile that minimizes the technical biases but still represents finer level of granularity than a bulk sample. In order to define cell groupings that would lead to functional interpretation we used a matrix factorization technique called ICA

ICA is a matrix factorization-based technique aiming at defining statistically independent hidden factors shaping gene expression. Stability-based analysis revealed only one sufficiently stable independent component in the case of both Macrophage and NK data subsets. Therefore, first independent component was used to rank the individual cells. We grouped the NK single cells depending on the first independent component (IC1) projection score such that Group 1 had positive projection scores and the Group 2 has negative projection scores. For macrophage single cells we selected the first and the last quartiles of the macrophage scores of IC1 projection. In order to best interpret the "extreme" tendencies of the cells placed on the opposite side of IC. The distinction of the groups plotted in first and the second principal components space (PC1 and PC2) can be seen in Figs. 4a and 5a.

For cell groups defined as described above, the following procedure was applied in order to define the map module scores. For each module, 50% of most variant genes were retained in order to select genes over the median variability. The module score was defined as the mean of the selected genes.

Standard t-test was used to assess statistical differences between single-cell groups for each module. The $p$ values of the $t$-test were reported in the heatmaps with the standard code of significance (***$p < 0.001$, **$p < 0.01$, *$p < 0.05$, <0.1).

The data on pan-cancer meta-analysis of expression signatures from ~18,000 human tumors across 39 malignancies accompanied by survival clinical data were used[48]. In total, 6323 genes with significant $z$-scores ($p$ value <0.05) indicating correlation to patient survival were retrieved[48] and overlapped with the gene lists from the innate immune response meta-map. Enrichment of the meta-map with the genes significantly positively or negatively correlated with patient survival was assessed using the $\chi^2$ test with $p$ value threshold 0.001.

**Reporting summary**. Further information on research design is available in the Nature Research Reporting Summary linked to this article.

## Data availability

The cell-type-specific maps and meta-map of innate immune response in cancer are freely available at the web page (https://navicell.curie.fr/pages/maps_innateimmune.html). The meta-map and cell-type-specific maps are provided in three platforms, NaviCell, MINEVRA and integrated into the repository NDEx. The maps exist and can be downloaded in several exchange formats (CellDesigner SBML level 2 version 4, SBGN-ML 0.2, SBML level 3 version 1, Cytoscape CX version 3.4.0). In addition, the composition of map signaling pathways, modules, and meta-modules is provided in a form of GMT files (Supplementary Tables 2 and 3, respectively) suitable for further

functional data analysis. A network of binary relations between proteins generated from the meta-map and the complete list of references annotating the maps are also available.

## Code availability

The documentation and the scripts for module activity calculation and generation of life example is provided at GitHub (https://github.com/sysbio-curie/NaviCell/tree/master/auxiliary_scripts). The step-by-step procedure on modular hierarchical maps construction is also provided at https://github.com/sysbio-curie/NaviCell.

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

## Acknowledgements

We thank Daniel Rovera for help with network structure analysis and L. Cristobal Monraz Gomez for help with data visualization and critical reading of the paper. We thank Marek Ostaszewski and Piotr Gawron for integration of the resource into the MINERVA platform. This work has been funded by INSERM Plan Cancer No. BIO2014-08 COMET grant under ITMO Cancer BioSys program. This work received support from MASTODON program by CNRS (project APLIGOOGLE), COLOSYS grant ANR-15-CMED-0001-04, provided by the Agence Nationale de la Recherche under the frame of ERACoSysMed-1, the ERA-Net for Systems Medicine in clinical research and medical practice and by IMI2-IMMUcan grant. ITMO cancer (AVIESAN) provided 3-year PhD grant and foundation Bettencourt Schueller and Center for Interdisciplinary Research supported the training of the Ph.D. student.

## Author Contributions

M.K. constructed signaling networks, performed data visualization, and wrote the paper; U.C. performed data analysis and enrichment calculations and wrote the paper; N.S. performed statistical analysis of maps content, integration of the resource into browsable platforms, and wrote the paper; S.D.A. and E.B. advised during the project and revised the paper; V.S. advised during the project and critically revised and restructured the paper; A.Z. supervised the data analysis, advised during the project, and revised the paper; and I.K. led the project and wrote the paper.

## Conflict of interest

The authors declare that they have no conflict of interest.
