## [Peer Review File · Nature Communications]

Reviewers' comments:

Reviewer #1 (Remarks to the Author):

The manuscript by Kondratova et al describes an online resource for visualizing signaling and network functions of innate immune cells in the context of the tumor microenvironment. The maps are curated from literature and other databases and are organized by innate immune cell type. The maps are also integrated into a "meta-map" that appears to visualize/quantify all of the signaling pathways activated across cell types to produce a view of how "pro-tumor" or "anti-tumor" the landscape is. After introducing the tool, the authors apply it to single-cell sequencing data from macrophages and NK cells in metastatic melanoma (Tirosh et al). They map pathway activation of individual cells to show that subpopulations are anti-tumor vs. pro-tumor.

Given the complexity and heterogeneity of innate immune cells in cancer, the overall idea is interesting and the web-based tool potentially useful. However, the overall description of the modules and calculations were very hard to understand. In particular, one of major contributions of this work is to provide users who are studying the innate immune cells in cancer an overall network-level view of how "anti-tumor" or "pro-tumor" the innate immune cells are. However, the description of the analyses that underlie that assessment need to be improved in order to evaluate if this web resource is accomplishing that goal.

Major comments

1. In the meta-map the authors state that the signaling is propagated to eventually define the overall influence on tumor growth (line 217), however it's unclear how that is accomplished. Are they hypothesizing outcomes based on pathway connections? If so, how is the role in tumor growth defined?
2. The authors report anti-tumor activity scores but I don't understand how these were calculated. In the Methods, a method for computing activity scores is described for the specific example from the Tirosh data. Is that also the general way that they are calculated? Why are 50% of the variable genes retained? Is the t-test of significance corrected for multiple comparisons (or am I misunderstanding how the t-test was applied)?
3. The application to the Tirosh data was an interesting aspect to the paper, however it was not a straightforward visualization of scRNA-seq data because the authors first preprocessed the data with ICA to define different groups and then looked at pathways in each group using their software. However, it's likely that most immunologists would simply load in the data. What kind of information would that return? Is it possible to arrive at similar conclusions that way?
4. I tried to run the "live example" on the Navicell web site but it always got stuck around the 4th step. Also, it would be helpful if an area of the map could be highlighted and all of the entities that are included in that module could be browsed. If that feature was there, I didn't see it.

Minor comments

1. Supp. Fig. 1 and line 124: authors refer to a confidence score for reactions, which would be useful, but I did not see that.
2. Line 333 is very confusing: "...one can see that the independent component computed are attracted by the bimodality characterizing the distribution." Can this be rewritten to more clearly state what they mean?

Reviewer #2 (Remarks to the Author):

Kondratova et al describe a large-scale molecular map of processes involved in the innate immune response in cancer, as well as its application to a dataset of single cell RNA-Seq data from macrophages and NK cells in metastatic melanoma. The final map is the result of integration of multiple cell-type specific maps which were constructed through manual literature curation and integration of data from other resources.

The work presented in the manuscript represents a very significant effort to collect and structure biomolecular knowledge around the processes of interest. The multi-layer approach from biological processes through modules to detailed pathways aims to structure the data and make it more comprehensible.

Unfortunately, the presentation of the map in the NaviCell tool is essentially unusable. After initial problems, I decided to do a systematic test. To mitigate against network bandwidth effects as well as potential problems in the hardware used, I did a comparative test: In different tabs of the same browser (tested both Safari and Chrome, latest released versions on Mac), at the same time, I loaded:

1: KEGG: https://www.genome.jp/kegg-bin/show_pathway?map04620

This is only for comparison, the map is small, and KEGG offers only limited interactivity in terms of panning and zooming. However, interaction was smooth, with good responsiveness.

2: Reactome: <https://reactome.org/PathwayBrowser/#/R-HSA-166658&PATH=R-HSA-168256,R-HSA-168249>

Reactome does not offer very large maps like the NaviCell one under consideration here. However, the chosen map has a considerable size and connectivity. Interaction with the selected map, as well as higher level pictorial representations, was smooth and interactive for panning, zooming, and object selection under Chrome. It was slower, but acceptable (up to ca. 3 secs delay) under Chrome.

3: Minerva: <https://vmh.uni.lu/#reconmap2>

This map, in terms of entities, though probably not connections, is larger than the NaviCell map. Panning, zooming, and object selection were all smooth and interactive, with responses almost immediate, or within very few seconds, not hampering exploration of the map.

4: NaviCell: https://navicell.curie.fr/navicell/newtest/maps/innate_immune/master/index.html

The tool, with the above map, as well as others referenced in the manuscript, is practically unusable. A simple panning operation with the mouse, from releasing the mouse button to completed rebuilt of the image on screen took 40 (!) seconds under Chrome, any other operation similarly has huge delays. Safari was more responsive, but still unusable with response times of around 10 seconds for simple panning operations. A systematic evaluation, let alone interactive discovery of the map, is essentially impossible; I did not do a detailed evaluation of the tool, as this was basically not possible.

Lesser concerns:

The authors compare their work to KEGG, Reactome, and InnateDB. However, I could not find any information on which database versions were used. This needs to be provided.

In addition to these three comparators, the authors list various databases as data sources in the Methods section. All of these should be listed with the version used, and the appropriate literature citation.

Reviewer #3 (Remarks to the Author):

The work of Kondratova et. al. describes a comprehensive multi-scale network maps of the innate immune system. The fully annotated maps were constructed manually from hundreds of

published papers, and consist of over a thousand of components. The maps, contextualized in cancer and several specific cell types, are available to the community in the NaviCell software, which enables others to interactively browse through the maps at the various levels of biological organization. Finally, the authors used the maps to visualize results of RNAseq data analyses that illustrated the existence of anti- and pro-tumor macrophage sub-populations in metastatic melanoma.

First, this work is impressive in the expansive and comprehensive nature of the multi-scale network maps. In particular, it is great to see that the authors have put quite a bit of effort to not only synthesize the map, but to also fully annotate it at a very high level detail (unlike many other existing maps and computational models that lack substantial annotations). Both, its large scope and annotation availability have the potential to significantly increase the use and reuse of the described resource. While I think this manuscript is suitable for Nature Communications, there are several issues with the manuscript as is. In particular, the manuscript suffers from quite a bit of grammar and sentence structure issues. Finally, there are several related aspects that I think would strengthen the manuscript, if they were discussed. Please see below.

Grammar/Typos/Sentence Structure Issues/Clarity of statements:

- please note that this is not meant to be an exhaustive list. I tried to point out as many issues as possible, but there were too many to catch them all through a limited number of readings.

- * (Line) #29 - "from" or "based on" seems to be extra, left over words; one of them should go
- * #29 - missing period at the end of the sentence
- * #50 - TGFB and IL10 should be spelled out as they are used for the first time
- * #50 - "TGFB, IL10 and growth factors" is confusing and redundant as TGF beta is a growth factor.
- * #60 - please spell out abbreviations when used for the first time (please check throughout the manuscript)
- * #101/102 - "data interpretations and modelling" seems very vague. While the manuscript does discuss the visualization of RNAseq data analysis on top of the maps, the manuscript provides very little guidance of how the map could be used for "modelling".
- * #109 - "normally" -- the word is confusing in this context (i.e., are you suggesting that there is an other than normal layout?); I would remove the word.
- * #112 - "map managers" -- I am not sure what that means? Is that referring to the people who constructed the map? It sounds like internal team jargon.
- * #129/130 - Isn't NaviCell also "a tool for interactive web-based data visualization"? What is the relationship/similarity/differences between NaviCell and ACSN? Unless more is to be provided to the reader about how ACSN fits in with this work, I would suggest removing the sentence.
- * #133 - the sentence doesn't seem correctly structured; should "as" become "include"? Perhaps split the sentence in two? I read the sentence several times, and I am still unsure what it says. :(
- * #175 - "which" -> "that"
- * #177 - should be "NK cell" (missing "cells")
- * #196 - extra space before "."
- * #196 - the first use of the term "meta-map" -- at this point of the manuscript, I am really not sure what to imagine under that term. I suggest adding a few descriptors (or another sentence) to better explain what that means.
- * #218 - "which" -> "that"
- * #218 - I am not sure what "The latest entities" means. I think the word "latest" is not used in the proper context here.
- * #222 - missing "," before "which transforms"
- * #231 - "especially relevant for structural analysis and modelling studies" is quite vague -- adding a few examples would be useful here.
- * #239 - please add "and" in "'Immune activation', and 'Tumor killing' processes"
- * #240 - missing "," before "and 'Tumore growth'"
- * #243 - missing "." at the end of the sentence

- * #255 - "on" -> "of"
- * #262 - missing "the" in "by the Google Maps engine"
- * #297 - missing "are" in "that are very well studied"
- * #314/315 - remove "is the one that"
- * #315 - incorrect use of "in opposite to"; probably this should be "in contrary to"
- * #319 - incorrect tense  "built-in"
- * #319 - "that" -> ", which"
- * #393 - missing "pathways" in "there are three pathways"
- * #398 - "which" -> "that"
- * #401 - remove "the"
- * #402 - remove "that"; "represents" -> "representing"
- * #404-406 - the sentence starting with "Interestingly .." seems structurally incorrect, and needs revised
- * #430 - "The last years" -> "Recent years"
- * #435 - "the set" -> "a set"
- * #442 - missing "the" in "of the immune system"
- * #443 - remove ",," in "the second, as an integrated"
- * #446 - "but not too controversial picture" is vague and unclear what the authors mean by that.
- * #452 - remove "the" from "into the pro-tumor and anti-tumor zones"
- * #455 - "that" -> "which"
- * #471 - missing "the" -> "Construction of the innate immune response map"
- * #471 - "the" -> "an" in "in the attempt to build.."
- * #472 - "as fibroblasts" -> "including fibroblasts"

Other comments/questions/suggestions for improvement

* While the comprehensive map will provide a great resource to the community as is, it is unclear what the authors' plan is continue to and/or allow the community to maintain, update, and grow the map. It would be great if the larger community could expand and update the map, however it is not clear whether that is possible. If it is possible, how will the authors manage and ensure the quality of the contributions to the map?

* Starting with line #303, the authors emphasize the small overlap of the literature used to construct their map, and other databases. I am not clear as to why a small overlap would be a good thing. Instead, I would expect that the literature used to construct related pathways in other databases would be largely included in the authors' used literature, indicating that they used and re-used existing pathway resources from the other databases, instead of ignoring them and building their own. I understand that the existing pathway resources may not be cell- and cancer-specific, but I can imagine there would be many existing pathways that fit the authors' specific context. I think the reasoning behind this analysis should be better explained in the manuscript.

Point-by-point response to comments

Reviewer #1 (Remarks to the Author):

The manuscript by Kondratova et al describes an online resource for visualizing signaling and network functions of innate immune cells in the context of the tumor microenvironment. The maps are curated from literature and other databases and are organized by innate immune cell type. The maps are also integrated into a “meta-map” that appears to visualize/quantify all of the signaling pathways activated across cell types to produce a view of how “pro-tumor” or “anti-tumor” the landscape is. After introducing the tool, the authors apply it to single-cell sequencing data from macrophages and NK cells in metastatic melanoma (Tirosh et al). They map pathway activation of individual cells to show that subpopulations are anti-tumor vs. pro-tumor.

Given the complexity and heterogeneity of innate immune cells in cancer, the overall idea is interesting and the web-based tool potentially useful. However, the overall description of the modules and calculations were very hard to understand. In particular, one of major contributions of this work is to provide users who are studying the innate immune cells in cancer an overall network-level view of how “anti-tumor” or “pro-tumor” the innate immune cells are. However, the description of the analyses that underlie that assessment need to be improved in order to evaluate if this web resource is accomplishing that goal.

We thank the reviewer for these remarks. We edited the text in order to clarify how the anti- and pro-tumor zones on the maps were created and visually represented:

First of all, the maps are organized during their construction such that the layout of each map contains the notion of “anti- and pro-tumor”. The content of the maps is organized around signalling pathways which compose functional modules. These modules are grouped into meta-modules that in turn occupy two zones: “pro-tumor” and “anti-tumor”. Therefore, the “geographical” location of biochemical processes on the maps correspond to the type of outcome, namely anti-tumor or pro-tumor. We explain on pp. 4-6 and corresponding Figure 2 how it was done for cell type-specific maps. We explain on pp. 6-7 and corresponding Figure 3 how it was done for the integrated innate immune meta-map.

Finally, we better describe the entities and reaction annotations. With this purpose, the STAR methods section was largely re-written and massively extended (pp.20-23). Among other points, it explains the map structure and tagging system, confidence score assignment, and NaviCell annotation format.

We also provide now the entire content of the map, namely signalling pathways, modules, meta-modules, anti- and pro-tumor zones in the form of downloadable tables as supplementary to this paper and as files downloadable from the website (http://navicell.curie.fr/pages/maps_innateimmune.html). These tables can be directly used for further functional analyses.

Major comments

1. In the meta-map the authors state that the signaling is propagated to eventually define the overall influence on tumor growth (line 217), however it's unclear how that is accomplished. Are they hypothesizing outcomes based on pathway connections? If so, how is the role in tumor growth defined?

We added into the STAR methods an explanation of the ‘phenotype’ nodes role in the map model (P. 21).

The essence of manual knowledge formalization in the form of networks is in that the information on molecular mechanisms is collected from multiple papers and manually placed together using standard syntax and according to the rules for process representation accepted in the systems biology field.

Indeed, some phenotypes (=biological outcomes, readouts) are governed by one pathway. However, some especially complex biological outcomes as Tumor growth, are regulated by multiple pathways. This defines a “functional module”, namely, a set of pathways that together lead to the outcome. For example, this is exactly the case of ‘Tumor growth’ functional module in the innate immune meta-map (see e.g. Figure 3B).

To make the message clearer, we added Supplementary Figure 2 that shows a signalling pathway (Suppl. Figure 2A) where a biological process is initiated by one molecule/complex (often a receptor/ligand) and the signal propagates downstream. We show in the Figure 2B an example of a functional module where several pathways converge, guiding the final phenotype (which can be also termed biological outcome or readout).

Detailed methodology of map creation, graphical syntax meaning and principles behind the map structuring are described with representative examples in the following paper that we cite in the manuscript on pp. 4 and 21:

Kondratova M, Sompairac N, Barillot E, Zinovyev A, Kuperstein I. Database (Oxford). 2018 Jan 1;2018. Signalling maps in cancer research: construction and data analysis doi:10.1093/database/bay036.

2. The authors report anti-tumor activity scores but I don't understand how these were calculated. In the Methods, a method for computing activity scores is described for the specific example from the Tirosh data. Is that also the general way that they are calculated?

We agree that the approach was not detailed enough. It is now detailed in the STAR methods section.

Indeed, the pro-tumor and anti-tumor activity scores, as well as scores for any other modules, are calculated for NK and Macrophages single cell subsets using gene expression data from Tirosh et al. The procedure is rather general and can be applied to any gene set, e.g. for the definition of modules from cell type-specific maps, meta-map as well as for pro- and anti-tumor zones.

The application examples aim to demonstrate the added value of manually created cell type-specific and integrated meta map in innate immune response in cancer from the latest literature. The well-thought structure and organization of the maps also allow meaningful visualization of module activity scores such that the interpretation of the data is easy and intuitive.

Why are 50% of the variable genes retained? Is the t-test of significance corrected for multiple comparisons (or am I misunderstanding how the t-test was applied)?

50% was chosen as threshold selecting genes above median variance value. However, after the remark from the reviewer we checked that in a large interval of thresholds our conclusions remain valid and the corresponding p-values remain significant (see the file “review_precisions_additives.pdf”)

The t-test was applied for each module in each map. As the number of modules is relatively small (19 in innate immune meta-map, 6 in Macrophages cell type-specific map and 7 in NK cell type-specific map) many methods for FDR control is difficult to apply directly. Therefore, simplest Bonferoni correction (a stringent solution for p-values correction which divides the p-value threshold by the number of modules) was used and modules marked with more than one asterix (** and ***) remain significant after such a correction. The modules that are not significant anymore are the ones of the Innate map NK cell: INTEGRINS, IMMUNOSTIMULATORY_CYTOKINE_EXPRESSION which does not change the overall conclusions presented in the article.

3. The application to the Tirosh data was an interesting aspect to the paper, however it was not a straightforward visualization of scRNA-seq data because the authors first preprocessed the data with ICA to define different groups and then looked at pathways in each group using their software. However, it's likely that most immunologists would simply load in the data. What kind of information would that return? Is it possible to arrive at similar conclusions that way?

Looking at single cell data “directly” is connected with a number of technical issues unlike bulk patient data where individual tumor samples can be individually visualized and analysed. A heterogeneity of single cell gene expression profiles can be predominantly of technical origin [<https://www.ncbi.nlm.nih.gov/pmc/articles/PMC5465644/>]. Therefore, application of various data processing techniques, usually introducing some kind of aggregation of single cell profiles or groups of genes, are being developed in order to conclude anything meaningful on the underlying biology [<https://journals.plos.org/ploscompbiol/article?id=10.1371/journal.pcbi.1006245>].

In our work, we use a relatively straightforward and standard way of treating normalized single cell data. Using ICA approach, we define a way to rank individual cells along axes that represent latent variables that might correspond to a biological function or a cell state. Our previous analyses of tumor bulk transcriptomes as well as works of many other groups proved that ICA gives biologically meaningful results in detecting signals shaping transcriptomes, e.g. as described in

Biton A, et al. Independent component analysis uncovers the landscape of the bladder tumor transcriptome and reveals insights into luminal and basal subtypes. Cell Rep. 2014, 9(4):1235-45. doi: 10.1016/j.celrep.2014.10.035.

It is worth mentioning that in several widely used pipelines of single cell RNASeq data analysis (such as Monocle) ICA is used as a standard dimensionality reduction step which is frequently more informative than PCA.

Subsequently, we compute module activity scores of map modules for cells aggregated in groups. The module activity scores based on the mean expression of module genes allow detection of driving tendencies smoothing the effect of confounding technical biases.

However, intrigued by the reviewer suggestion, we plotted the gene expression values of randomly selected 42 NK cells in the NK specific Immune Map (see “review_precision_additives.pdf”). It is not trivial for us to make global conclusion while visualizing separately each single cell in the context of innate immune meta-map. It can be observed that some cells indeed approach the average phenotypes of the two groups that we define in this work. However, no meaningful conclusion can be drawn from just loading the data of each cell, as example shows. This explains and justifies the chosen approach to define cell groups using ICA method and thus calculate the activities scores for these groups.

4. I tried the to run the “live example” on the Navicell web site but it always got stuck around the 4th step. Also, it would be helpful if an area of the map could be highlighted and all of the entities that are included in that module could be browsed. If that feature was there, I didn't see it.

We repaired this function.

The feature of selection is not available in NaviCell. However, there is a feature where it is possible to select neighbours of a certain entity and highlight them. The MINERVA platform allows selecting a region of the map by drawing a shape and exporting this region as an image or map. NDEx has a feature of selection similar to Cytoscape where it is possible to select entities and highlight them. It is also possible to download GMTs files

containing list of genes related to different levels of the map, such as Modules, Metamodules or Signalling Pathways
(https://navicell.curie.fr/navicell/newtest/maps/innate_immune/master/Innate_Immunity_Response_GMTs.zip)

Minor comments

1. Supp. Fig. 1 and line 124: authors refer to a confidence score for reactions, which would be useful, but I did not see that.

The description of confidence scores is now added into the STAR methods (p.22) and the annotation example showing confidence score is provided in Supplementary figure 1.

2. Line 333 is very confusing: "...one can see that the independent component computed are attracted by the bimodality characterizing the distribution." Can this be rewritten to more clearly state what they mean?

Thank you for this comment, we modified the analysis description in the manuscript.

Reviewer #2 (Remarks to the Author):

Kondratova et al describe a large-scale molecular map of processes involved in the innate immune response in cancer, as well as it's application to a dataset of single cell RNA-Seq data from macrophages and NK cells in metastatic melanoma. The final map is the result of integration of multiple cell-type specific maps which were constructed through manual literature curation and integration of data from other resources.

The work presented in the manuscript represents a very significant effort to collect and structure biomolecular knowledge around the processes of interest. The multi-layer approach from biological processes through modules to detailed pathways aims to structure the data and make it more comprehensible.

Thank you for this comment.

Unfortunately, the presentation of the map in the NaviCell tool is essentially unusable. After initial problems, I decided to do a systematic test. To mitigate against network bandwidth effects as well as potential problems in the hardware used, I did a comparative test: In different tabs of the same browser (tested both Safari and Chrome, latest released versions on Mac), at the same time, I loaded:

1: KEGG: https://www.genome.jp/kegg-bin/show_pathway?map04620

This is only for comparison, the map is small, and KEGG offers only limited interactivity in terms of panning and zooming. However, interaction was smooth, with good responsiveness.

2: Reactome: <https://reactome.org/PathwayBrowser/#/R-HSA-166658&PATH=R-HSA-168256,R-HSA-168249>
Reactome does not offer very large maps like the NaviCell one under consideration here. However, the chosen map has a considerable size and connectivity. Interaction with the selected map, as well as higher level pictorial representations, was smooth and interactive for panning, zooming, and object selection under Chrome. It was slower, but acceptable (up to ca. 3 secs delay) under Chrome.

3: Minerva: <https://vmh.uni.lu/#reconmap2>

This map, in terms of entities, though probably not connections, is larger than the NaviCell map. Panning, zooming, and object selection were all smooth and interactive, with responses almost immediate, or within very few seconds, not hampering exploration of the map.

4: NaviCell: https://navicell.curie.fr/navicell/newtest/maps/innate_immune/master/index.html

The tool, with the above map, as well as others referenced in the manuscript, is practically unusable. A simple panning operation with the mouse, from releasing the mouse button to completed rebuilt of the image on screen took 40 (!) seconds under Chrome, any other operation similarly has huge delays. Safari was more responsive, but still unusable with response times of around 10 seconds for simple panning operations. A systematic evaluation,

let alone interactive discovery of the map, is essentially impossible; I did not do a detailed evaluation of the tool, as this was basically not possible.

We appreciate the systematic approach of the reviewer and thank for dedicating time and energy to this. We are also sorry about the technical problems encountered.

First, we have to stress that slow performances of NaviCell platform was observed during the paper revision period due to unfortunate coincidence: Google Maps have moved at this time to a different system and since NaviCell is using Google Maps API, it affected the performance. This problem does not exist anymore.

The comparison exercise performed by the reviewer inspired us to perform integration of innate immune response resource into additional platforms. Now all maps from the resource are also available under MINERVA and Ndex platforms and accessible for the resource website (http://navicell.curie.fr/pages/maps_innateimmune.html).

Overall, we would like to stress that the manuscript is devoted to the description of the innate immune in cancer map as a resource, while NaviCell is one possible way to present it to the users. In order to provide smooth access to the map content in the most convenient way, we are very open to use alternative pathway browsing platforms able to import CellDesigner maps in one way or another.

Lesser concerns:

The authors compare their work to KEGG, Reactome, and InnateDB. However, I could not find any information on which database versions were used. This needs to be provided.

The database versions were explicitly added into the manuscript.

In addition to these three comparators, the authors list various databases as data sources in the Methods section. All of these should be listed with the version used, and the appropriate literature citation.

Versions and citations were added.

Reviewer #3 (Remarks to the Author):

The work of Kondratova et. al. describes a comprehensive multi-scale network maps of the innate immune system. The fully annotated maps were constructed manually from hundreds of published papers, and consist of over a thousand of components. The maps, contextualized in cancer and several specific cell types, are available to the community in the NaviCell software, which enables others to interactively browse through the maps at the various levels of biological organization. Finally, the authors used the maps to visualize results of RNAseq data analyses that illustrated the existence of anti- and pro-tumor macrophage sub-populations in metastatic melanoma.

First, this work is impressive in the expansive and comprehensive nature of the multi-scale network maps. In particular, it is great to see that the authors have put quite a bit of effort to not only synthesize the map, but to also fully annotate it at a very high level detail (unlike many other existing maps and computational models that lack substantial annotations). Both, its large scope and annotation availability have the potential to significantly increase the use and reuse of the described resource.

We appreciate a lot this positive attitude of the reviewer.

While I think this manuscript is suitable for Nature Communications, there are several issues with the manuscript as is. In particular, the manuscript suffers from quite a bit of grammar and sentence structure issues. Finally, there are several related aspects that I think would strengthen the manuscript, if they were discussed. Please see below.

Grammar/Typos/Sentence Structure Issues/Clarity of statements:

The paper was extensively revised by English-speaking persons.

- please note that this is not meant to be an exhaustive list. I tried to point out as many issues as possible, but there were too many to catch them all through a limited number of readings.

We included all these changes.

- * (Line) #29 - "from" or "based on" seems to be extra, left over words; one of them should go
- * #29 - missing period at the end of the sentence
- * #50 - TGFB and IL10 should be spelled out as they are used for the first time
- * #50 - "TGFB, IL10 and growth factors" is confusing and redundant as TGF beta is a growth factor.
- * #60 - please spell out abbreviations when used for the first time (please check throughout the manuscript)
- * #101/102 - "data interpretations and modelling" seems very vague. While the manuscript does discuss the visualization of RNAseq data analysis on top of the maps, the manuscript provides very little guidance of how the map could be used for "modelling".
- * #109 - "normally" -- the word is confusing in this context (i.e., are you suggesting that there is an other than normal layout?); I would remove the word.
- * #112 - "map managers" -- I am not sure what that means? Is that referring to the people who constructed the map? It sounds like internal team jargon.
- * #129/130 - Isn't NaviCell also "a tool for interactive web-based data visualization"? What is the relationship/similarity/differences between NaviCell and ACSN? Unless more is to be provided to the reader about how ACSN fits in with this work, I would suggest removing the sentence.
- * #133 - the sentence doesn't seem correctly structured; should "as" become "include"? Perhaps split the sentence in two? I read the sentence several times, and I am still unsure what it says. :(
- * #175 - "which" -> "that"
- * #177 - should be "NK cell" (missing "cells")
- * #196 - extra space before "."
- * #196 - the first use of the term "meta-map" -- at this point of the manuscript, I am really not sure what to imagine under that term. I suggest adding a few descriptors (or another sentence) to better explain what that means.
- * #218 - "which" -> "that"
- * #218 - I am not sure what "The latest entities" means. I think the word "latest" is not used in the proper context here.
- * #222 - missing "," before "which transforms"
- * #231 - "especially relevant for structural analysis and modelling studies" is quite vague -- adding a few examples would be useful here.
- * #239 - please add "and" in "'Immune activation', and 'Tumor killing' processes"
- * #240 - missing "," before "and 'Tumore growth'"
- * #243 - missing "." at the end of the sentence
- * #255 - "on" -> "of"
- * #262 - missing "the" in "by the Google Maps engine"
- * #297 - missing "are" in "that are very well studied"
- * #314/315 - remove "is the one that"
- * #315 - incorrect use of "in opposite to"; probably this should be "in contrary to"
- * #319 - incorrect tense  "built-in"
- * #319 - "that" -> ", which"
- * #393 - missing "pathways" in "there are three pathways"

- * #398 - "which" -> "that"
- * #401 - remove "the"
- * #402 - remove "that"; "represents" -> "representing"
- * #404-406 - the sentence starting with "Interestingly .." seems structurally incorrect, and needs revised
- * #430 - "The last years" -> "Recent years"
- * #435 - "the set" -> "a set"
- * #442 - missing "the" in "of the immune system"
- * #443 - remove "," in "the second, as an integrated"
- * #446 - "but not too controversial picture" is vague and unclear what the authors mean by that.
- * #452 - remove "the" from "into the pro-tumor and anti-tumor zones"
- * #455 - "that" -> "which"
- * #471 - missing "the" -> "Construction of the innate immune response map"
- * #471 - "the" -> "an" in "in the attempt to build.."
- * #472 - "as fibroblasts" -> "including fibroblasts"

Other comments/questions/suggestions for improvement

* While the comprehensive map will provide a great resource to the community as is, it is unclear what the authors' plan is to continue to and/or allow the community to maintain, update, and grow the map. It would be great if the larger community could expand and update the map, however it is not clear whether that is possible. If it is possible, how will the authors manage and ensure the quality of the contributions to the map?

We added a paragraph into the paper text (p. 14), discussing this important point.

The integration of the innate immune response in cancer map into other online map browsing platforms allows broader exposure and use of the map. Therefore, in addition to NaviCell platform, the resource is also exposed in the MINERVA platform and integrated into the NDEX repository and platform. In the future, the resource will be also integrated into big collections as Pathway Commons, Wiki Pathways, etc. These moves will allow a deeper involvement of the scientific community into the maintenance and update of the maps with the latest discoveries.

The maps in the resource can be edited by the community under the various platforms. The creators of the innate immune resource will carefully evaluate the suggestions and include valuable ones into the 'master' version of the map which will be updated through all online platforms where it will be hosted.

* Starting with line #303, the authors emphasize the small overlap of the literature used to construct their map, and other databases. I am not clear as to why a small overlap would be a good thing. Instead, I would expect that the literature used to construct related pathways in other databases would be largely included in the authors' used literature, indicating that they used and re-used existing pathway resources from the other databases, instead of ignoring them and building their own. I understand that the existing pathway resources may not be cell- and cancer-specific, but I can imagine there would be many existing pathways that fit the authors' specific context. I think the reasoning behind this analysis should be better explained in the manuscript.

We should underline that in the process of map construction, we did exploit the content of existing pathway databases in terms of the definition of pathway structures and pathway members. However, our goal was to annotate, revise and enrich existing pathway definitions (or introduce missing definitions) with information specific to *cancer biology* field and specific to *cell type*. In our objective, each mechanism shown in the maps should be connected to cancer progression and related to a specific cell type, which was not the purpose of wider scope pathway resources.

Therefore, small overlap with other generic databases confirms that the meta-map is not a mechanical compilation of existing databases, but rather an independent resource created having a specific focus in mind. It formalizes the part of biological knowledge which was not annotated before and highlights the difference between reconstruction of “generic” and cell-type specific pathways in terms of literature sources.

Another note is that in general crossing annotations of similar pathways in different databases typically leads to a relatively small overlap, indicating different models of literature curation. For example, some databases contain much more frequently reviews in their annotations while others contain more original experimental papers, where the mechanisms were first characterized. The total literature corpus about pathways is enormous and any pathway database use only a small sample of it, which typically leads to a small overlap.

The median age of the literature references in the meta-map is only one year younger compared to InnateDB and REACTOME. Nevertheless, there is a 27% of papers dated 2010-2017 in the literature body annotating the meta-map. It must be stressed that the topic of innate immune response specifically in cancer is rather young. Little was published in this field before 2010, therefore the papers composing the major part of the literature body used for Innate immune resource simply did not exist when the corresponding pathways in InnateDB and REACTOME were created.

Together, this explains the (expected!) difference in the literature coverage between innate immune response meta-map to InnateDB and REACTOME.

However, there is a need to demonstrate the coherence with the field. For this purpose, we included the confidence scores for entities and reaction on the maps (pp. 4 and 21, Supplementary figure 1).

Rev #1 Question 2

Study of change in the threshold of the percentage of variable genes chosen to compute the modules score.

A

MODULE	100%	90%	80%	70%	60%	50%	40%	30%	20%	10%
ANTIGEN_PRESENTATION_AND_IMMUNOSTIMULATORY_CHEKPOINTS	0	0	0	0	0	0	0	0	0	0
RECRUITMENT_OF_IMMUNE_CELLS	0,1725	0,1725	0,235	0,3073	0,8682	0,4608	0,0227	0,0001	0,0001	0,0001
MACROPHAGE	0,0692	0,0827	0,1151	0,1298	0,1798	0,2422	0,1605	0,0504	0,0035	0
IMMUNOSUPPRESSIVE_CYTOKINE_PATHWAYS	0	0	0,0001	0,0001	0,0007	0,0032	0,0028	0,0001	0	0
NO_ROS_PRODUCTION	0,1213	0,1118	0,1082	0,0683	0,0476	0,0692	0,0825	0,1331	0,3362	0,268
CORE_SIGNALING_PATHWAYS	0	0	0,0001	0,0001	0	0	0,0002	0,0001	0,0001	0
IMMUNOSTIMULATORY_CYTOKINE_PATHWAYS	0,0392	0,0489	0,0444	0,0318	0,1423	0,1819	0,0755	0,027	0,0226	0,0065
IMMUNOSTIMULATORY_CYTOKINE_EXPRESSION	0,136	0,1251	0,1327	0,1032	0,1037	0,1403	0,184	0,6563	0,2806	0

B

A- Table illustrating the change of p-value depending on the percentage of variable genes selected for the scores computation. We used as an example the NK-specific map but conclusions of this experiment are true for all maps and cell types. First, we defined minimal number of genes independent from the variability threshold set to 3 (for example if total number of genes in the module is 10, with 10% we would compute significance based on 1 gene, which is not statistically meaningful, in that case 3 most variant genes were retained for computation, in the case where there are 100 genes in the module, with 10% threshold, the score is computed for 10 genes). Then, we computed the p-values for each threshold varying from 10% (10% most variant genes) to 100% (all genes).

In the manuscript we selected the threshold of 50% in order to select genes above median variance value. It can be noticed that the modules significant at 50% are significant at all thresholds.

B – Heatmaps illustrating the module scores for groups 1 and 2 computed with different threshold of most variant genes. It can be observed that not only the differences observed at 50% threshold remain significant but also that the activated (red) modules remain activated and the non-activated (green) remain non-activated across the different thresholds.

Rev #1 Question 3

Visualizing the individual NK single cells with the NK-specific map

The maps proposed in the article enable a visualization of a sample which in the case of single cell transcriptomics correspond to a profile of a single cell. We selected randomly 12 single NK cells and we plotted their values on map without applying variability threshold (color correspond to the mean of gene expression values for genes present in the module). A high variability can be observed between the single cells.

Reviewers' comments:

Reviewer #1 (Remarks to the Author):

The manuscript by Kondratova et al describes an online resource for visualizing signaling and network functions of innate immune cells in the context of the tumor microenvironment. The maps are curated from literature and other databases and are organized by innate immune cell type. The maps are also integrated into a "meta-map" that visualizes all of the signaling pathways activated across cell types to produce a view of how "pro-tumor" or "anti-tumor" the landscape is. The authors apply their tool to single-cell sequencing data from macrophages and NK cells in metastatic melanoma (Tirosh et al). They map pathway activation of individual cells to show that subpopulations are anti-tumor vs. pro-tumor.

Given the complexity and heterogeneity of innate immune cells in cancer, the overall idea is interesting. In this revision, the authors have improved the explanations underlying their methodology. In particular, the method to provide users an overall network-level view of how "anti-tumor" or "pro-tumor" the innate immune cells are is improved. I agree with Reviewer 3, "other point" #1, that keeping the maps updated and therefore relevant will be a challenge, and I appreciate that the authors have added some text to address this. However, I think the usability of the map itself remains an issue.

Specific comments

1. In Fig. 1, step 6 indicates that the meta-map contains the influence of cell-cell interactions. How were these constructed? Also, the authors state that "neutrophils and mast cells are less studied and the molecular mechanisms implicated in the regulation of these cell types in TME are limited" therefore they only include the influence of these cells in the meta-map of innate immune response in cancer. Does it make sense to even include them there if the available data is so limited?

2. The most significant remaining issue is the usability of the map itself. The live example is working now, but it loads ovarian cancer data and then as it moves on, analyzes copy number data and mutation data until the map is very crowded and the images do not look like the figures presented in the paper. It would be more helpful to have the live example follow exactly what is reported in the manuscript, including the Tirosh data set as the example. I think this would significantly improve the chances that users would follow up with their own data analysis.

Minor comments

On line 339: "The innate immune response in cancer resource contains..." "in contrary" should be changed to "in contrast".

Typo in Fig. 1, step 6: ADDITIONAL NEUTROPHIL AND MAST CELL INTRACELLULAR INTERRACTIONS

Reviewer #2 (Remarks to the Author):

The authors have addressed my major concerns, the Navicell tool is now functional, and I appreciate the broader dissemination of the maps through MINERVA and NDEx. The text has significantly gained in clarity, too.

However, there are still some bugs and minor points:

On https://navicell.curie.fr/pages/maps_innateimmune.html the link to MINERVA results in an error:

Unexpected error occurred:

Unexpected token < in JSON at position 0

(Tested on latest Chrome on Mac)

As this is presumably outside the control of the authors, I don't see this as a critical shortcoming.

Is only the integrated map (link above) available in MINERVA and NDEx, or also the component maps for specific cell types? The first paragraph on P23 seems to state all, but the nice "landing page" with links to multiple platforms seems to be given only for the integrated map?

P3, L96: "...[in] KEGG and REACTOME signalling is represented in a patched manner, lacking cross-regulatory links and integrated presentation of multi-cellular system...": This statement is rather strong and should either be substantiated with objective criteria or rephrased. KEGG has e.g. https://www.kegg.jp/kegg-bin/highlight_pathway?scale=1.0&map=map04650&keyword=innate%20immune%20system and Reactome has very extensive crosslinks between pathways of the innate immune system.

P9 L306: It is not quite clear what the overlap numbers mean here. Three separate Venn diagrams as a supplemental figure, or a small table might help.

P10 L342: ..dues..

P12 L414: ..where.. -> were

Figure 1, section "Organisation", right hand side: "up to bottom" -> top to bottom ?

Reviewer #3 (Remarks to the Author):

Thanks to the authors for thoughtfully addressing my comments and concerns (as well as those of other reviewers).

Point-by-point response to comments after second revision

REVIEWERS

Reviewer #1 (Remarks to the Author):

The manuscript by Kondratova et al describes an online resource for visualizing signaling and network functions of innate immune cells in the context of the tumor microenvironment. The maps are curated from literature and other databases and are organized by innate immune cell type. The maps are also integrated into a “meta-map” that visualizes all of the signaling pathways activated across cell types to produce a view of how “pro-tumor” or “anti-tumor” the landscape is. The authors apply their tool to single-cell sequencing data from macrophages and NK cells in metastatic melanoma (Tirosh et al). They map pathway activation of individual cells to show that subpopulations are anti-tumor vs. pro-tumor.

Given the complexity and heterogeneity of innate immune cells in cancer, the overall idea is interesting. In this revision, the authors have improved the explanations underlying their methodology. In particular, the method to provide users an overall network-level view of how “anti-tumor” or “pro-tumor” the innate immune cells are is improved.

Thanks for this comment.

I agree with Reviewer 3, “other point” #1, that keeping the maps updated and therefore relevant will be a challenge, and I appreciate that the authors have added some text to address this.

Thanks for this comment.

We would like to stress that today indeed the map needs to be updated manually. However, we are starting a new project on integration of semi-automatic maps updates using text mining techniques in combination with web-based alert system. In short, the approach will allow the map managers to receive information on newly-published papers related to particular areas of the map (e.g. using map tags on modules, processes, cell type-specific tags, proteins/gene names, phenotypes names). After critical reading of the papers from these suggested literature sets, the map managers will update the map with new information.

This semi-automation of map content update is at the early stage, therefore we did not mention it in the paper text. However, we would like to reassure the reviewer that we are aware of the maps update issues and indeed address them.

Furthermore, we intend to integrate the map into resources as Pathway Commons, Wiki Pathways, GARUDA platform, etc. In part of these resources community-based curation is possible. This will be an additional map update mode.

However, I think the usability of the map itself remains an issue.

We addressed the usability issues of the maps in two ways:

- The first is rather technical as accessibility, supporting documentation, exploratory platforms, multiple formats available, etc.
- The second is rather conceptual, we mention in the discussion various application examples of similar efforts in different studies (in addition to our applications that we demonstrate in this paper).

These two improvements, we hope, answer to the reviewer’s question.

A. Live example:

We updated the live example, using Tirosh data such that it demonstrates how the figures from the manuscript can be produced. The data set used in the paper and in live example is also available in ‘Downloads’ on the map home page.

Moreover, the scripts for creation of live example and the documentation are provided at GitHub: https://github.com/sysbio-curie/NaviCell/tree/master/auxiliary_scripts

B. Procedure of maps generation:

Creation of maps with modular hierarchical structure as it is done for innate immune map, might be of interest for the readers developing similar approach. We would like to share our experience with the readers and therefore we provide the documentation and scripts under NaviCell, ‘Map pre-processing’ <https://github.com/sysbio-curie/NaviCell>

C. Exposure under different platforms:

Maps are now installed under three different platforms (NaviCell, NDeX, MINERVA) that actually represent the state-of-the-art tools for navigating complex maps. These are widely used platforms with different and complementary sets of features for navigation and map analysis. In this way maps can be better and widely exploited.

Now all maps, including comprehensive and cell type-specific maps, are exposed under the three web-platforms.

We would like to stress that in the future the map will be integrated into resources as Pathway Commons, Wiki Pathways, GARUDA platform, etc. This will assure dissemination of the resource and also increase usability due to different analytical features of these resources.

D. Multiple downloadable materials and maps in different formats:

We enriched the ‘Downloads’ with these materials that can be used for various studies https://navicell.curie.fr/pages/maps_innateimmune.html

For Enrichment analyses, map related gene signatures (modules, signalling pathways, etc.) can be downloaded here.

The Protein-Protein Interaction network (SIF) can be downloaded here.

The references for articles used for the construction of this map can be downloaded here.

The data used in the Live Example can be downloaded here. The data for the Map Staining was generated using the Module_Staining script available at GitHub

E. Application examples

We added to the discussion part examples of various applications of similar efforts (see page 14, last paragraph). This demonstrated the usability spectrum.

Specific comments

1. In Fig. 1, step 6 indicates that the meta-map contains the influence of cell-cell interactions. How were these constructed?

We thank the reviewer for drawing our attention to this point.

This expression is now replaced by more proper ‘CELL TYPE-SPECIFIC TAGS’. The idea of the integrated map is to represent innate immune signalling in a form of seamless total network. The cell type specificity tags of each entity on the map are inherited from the cell type-specific maps. These tags are assigned in the annotation of the entities.

Obviously, there are entities that found in several cell types. When cell type-specific maps are integrated together, entity's annotation are also merged together and therefore all cell-specific tags are preserved for a given entity.

In this paper we aimed to represent the integrated resource of signalling in innate immunity. We did not intend to explicitly represent inter cellular interactions between innate immunity cells.

Actually, representing it only at the level of innate immune response make little sense and would be very partial and incomplete. We would like to mention our ongoing project, aimed at a larger scale that describe not only innate immune players, but all components of tumor microenvironment. The cell type-specific tags (innate immune and other tumor microenvironment component's tags) will be definitely used in order to retrieve inter-cellular interactions. For example, the ligands are often present on one cell type and the corresponding receptors are found on another cell type. This information is presented in cell type-specific tag inside the corresponding annotations of these molecules on the maps. This is an example of inter-cellular type of interactions that will be systematically retrieved from maps. We intend to demonstrate inter-cellular influences between all tumor microenvironment components and tumor cell. However, this project is not in the scope of the current publication and will make a subject of a separate manuscript.

Also, the authors state that neutrophils and mast cells are less studied and the molecular mechanisms implicated in the regulation of these cell types in TME are limited" therefore they only include the influence of these cells in the meta-map of innate immune response in cancer. Does it make sense to even include them there if the available data is so limited?

The aim of our map is to represent the most complete and up-to-date encyclopaedia of molecular knowledge in innate immune response in cancer. As any project of this type, it is growing and evolving resource. We are convinced that the role of neutrophils and mast cells in cancer is very important and excluding them would be incorrect with respect to 'biological truth' in a way. The map will be enriched and update in the future, as new discoveries on the mechanisms of these cells in cancer will be published.

In addition, processes specifically representing mast cells and neutrophils are well matched with common pathways on the map that are present in all innate cell types. This completeness (limited by available information, of course) is useful for data analysis from these rare cell subsets in the context of our integrated map.

2. The most significant remaining issue is the usability of the map itself. The live example is working now, but it loads ovarian cancer data and then as it moves on, analyzes copy number data and mutation data until the map is very crowded and the images do not look like the figures presented in the paper. It would be more helpful to have the live example follow exactly what is reported in the manuscript, including the Tirosh data set as the example. I think this would significantly improve the chances that users would follow up with their own data analysis.

The suggestion of the reviewer to make the live example using Tirosh data from the paper actually inspired us not only to modify the live example, but also to provide to the users the procedure for creating their own live examples for their maps and using their datasets. With this aim, we exposed the documentation and the script for Live example generation. It is found in GitHub repository, we explicitly mention it in the materials and methods section of the paper.

Minor comments

On line 339: "The innate immune response in cancer resource contains..." "in contrary" should be changed to "in contrast".

Corrected

Typo in Fig. 1, step 6: ADDITIONAL NEUTROPHIL AND MAST CELL INTRACELLULAR INTERRACTIONS

The text in the Step 6 in Figure 6 has been changed (see answer to Specific comments #1)

Reviewer #2 (Remarks to the Author):

The authors have addressed my major concerns, the Navicell tool is now functional, and I appreciate the broader dissemination of the maps through MINERVA and NDEx. The text has significantly gained in clarity, too.

Thanks

However, there are still some bugs and minor points:

On https://navicell.curie.fr/pages/maps_innateimmune.html

the link to MINERVA results in an error:

Unexpected error occurred:

Unexpected token < in JSON at position 0

(Tested on latest Chrome on Mac)

As this is presumably outside the control of the authors, I don't see this as a critical shortcoming.

Corrected. Now MINERVA platform is fully operational.

Is only the integrated map (link above) available in MINERVA and NDEx, or also the component maps for specific cell types? The first paragraph on P23 seems to state all, but the nice "landing page" with links to multiple platforms seems to be given only for the integrated map?

All maps, integrated and cell type-specific are now accessible via three platforms: NaviCell, MINERVA and NDEx.

P3, L96: "...[in] KEGG and REACTOME signalling is represented in a patched manner, lacking cross-regulatory links and integrated presentation of multi-cellular system...": This statement is rather strong and should either be substantiated with objective criteria or rephrased. KEGG has e.g. https://www.kegg.jp/kegg-bin/highlight_pathway?scale=1.0&map=map04650&keyword=innate%20immune%20system and Reactome has very extensive crosslinks between pathways of the innate immune system.

We agree with the reviewer that this statement is inappropriate. We rephrased it, emphasising the importance of cancer-specific resource.

P9 L306: It is not quite clear what the overlap numbers mean here. Three separate Venn diagrams as a supplemental figure, or a small table might help.

The Venn diagram allow to grasp in an integrated manner the overlaps between number of publications from different databases. It shows that there are 785 unique publications used in meta-map. There are only several tenth of publications that are actually common between all databases, indicating uniqueness of each one of those.

We do think that with better explanation this figure is actually meaningful and interpretable. We explained this point more clearly in the corresponding figure legend (Supplementary figure 4A).

P10 L342: ..dues..

P12 L414: ..where.. -> were

Figure 1, section "Organisation", right hand side: "up to bottom" -> top to bottom ?

Corrected

Reviewer #3 (Remarks to the Author):

Thanks to the authors for thoughtfully addressing my comments and concerns (as well as those of other reviewers).

Thanks, we are glad the reviewer thinks that we succeeded to improve the paper in a proper way.

REVIEWERS' COMMENTS:

Reviewer #1 (Remarks to the Author):

The authors have thoroughly addressed my concerns. In particular, the live example is much easier to follow now and is a useful resource to accompany the manuscript.